# MIM-Refiner: A Contrastive Learning Boost from Intermediate Pre-Trained Masked Image Modeling Representations

**Benedikt Alkin**[1,2,@]  **Lukas Miklautz**[4]  **Sepp Hochreiter**[1,2,3]  **Johannes Brandstetter**[1,2]

[1]ELLIS Unit Linz, Institute for Machine Learning, JKU Linz, Austria
[2]Emmi AI GmbH, Linz, Austria    [3]NXAI GmbH, Linz, Austria
[4]Faculty of Computer Science, University of Vienna, Vienna, Austria
[@]alkin@ml.jku.at

## Abstract

We introduce MIM (Masked Image Modeling)-Refiner, a contrastive learning boost for pre-trained MIM models. MIM-Refiner is motivated by the insight that strong representations within MIM models generally reside in intermediate layers. Accordingly, MIM-Refiner leverages multiple instance discrimination (ID) heads that are connected to different intermediate layers. In each head, a nearest neighbor ID objective constructs clusters that capture semantic information which improves performance on downstream tasks, including off-the-shelf and fine-tuning settings.

The refinement process is short and simple – yet highly effective. Within a few epochs, we refine the features of MIM models from subpar to state-of-the-art, off-the-shelf features. Refining a ViT-H, pre-trained with data2vec 2.0 on ImageNet-1K, sets a new state-of-the-art in linear probing (84.7%) and low-shot classification among models that are pre-trained on ImageNet-1K. MIM-Refiner efficiently combines the advantages of MIM and ID objectives, enabling scaling ID objectives to billion parameter models using relatively little compute. MIM-Refiner compares favorably against previous state-of-the-art SSL models on various benchmarks such as low-shot classification, long-tailed classification and semantic segmentation. Project page: https://ml-jku.github.io/MIM-Refiner

## 1 Introduction

Self-supervised learning (SSL) has attracted considerable attention, owing to its data efficiency and generalization ability (Liu et al., 2021). SSL leverages pre-training tasks and creates intricate input representations without the need for explicit supervision via expensive annotations. In computer vision, Masked Image Modeling (MIM) (Chen et al., 2020b; Vincent et al., 2010; Pathak et al., 2016) has emerged as one of the prevalent SSL pre-training paradigms, enabling an efficient pretraining of large models on unlabeled data by reconstructing masked parts of the input images.

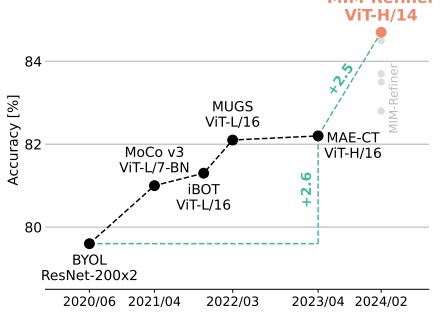

Figure 1: Linear probing state-of-the-art on ImageNet-1K over the last four years.

The success of MIM is driven by methods like Masked Autoencoder (MAE) (He et al., 2022), data2vec (Baevski et al., 2022; 2023), and others (Bao et al., 2022; Xie et al., 2022). For example, MAE opens the door for sparse pre-training of Vision Transformers (ViTs) (Dosovitskiy et al., 2021) by masking large parts of the image and not processing the masked areas. The computational efficiency, coupled with the data efficiency of a generative reconstruction task (Xie et al., 2023; El-Nouby et al., 2021; Singh et al., 2023) fosters the scaling to larger architectures on datasets of limited size. However, MIM models tend to spread their attention across the whole image (Walmer et al., 2023). When adapting to downstream tasks, a sufficient amount of labels is required to rewire the attention to focus on important regions in the image. In the absence thereof, MIM models perform

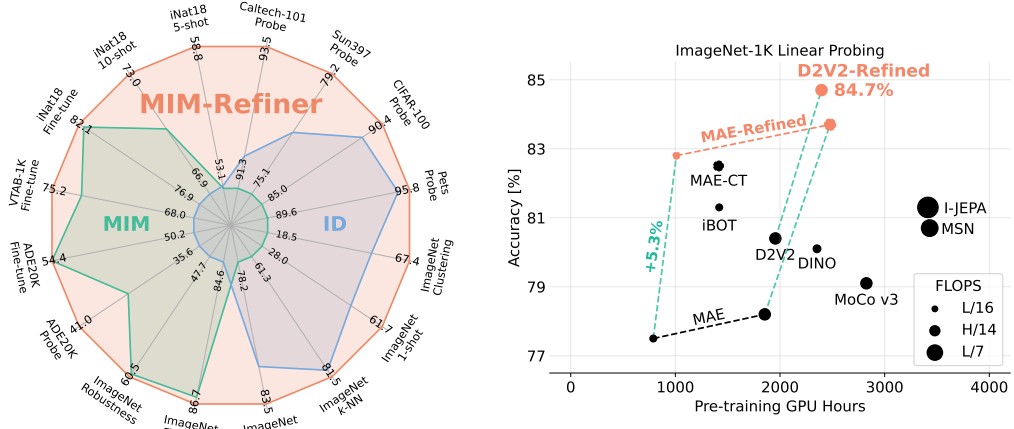

Figure 2: **Left:** Downstream evaluations of pre-trained SSL models. MIM-Refiner effectively combines their respective advantages of MIM and ID without suffering from their respective disadvantages. **Right:** Representation quality of SSL methods evaluated via linear probing. MIM-Refiner advances the state-of-the-art on ImageNet-1K pre-training in low- and high-compute regimes.

poorly. In contrast, for example, Instance Discrimination (ID) (He et al., 2020; Chen et al., 2020c) methods implicitly focus on objects and form semantic clusters in the latent space (Caron et al., 2021), which eases adaption to downstream tasks in the absence of vast amounts of labels. In summary, the most important desiderata for efficient SSL pre-training methods in computer vision are rich representations of the input – ideally in the form of semantic clusters in the latent space – alongside efficiency in both compute and data, and, most notably, favorable scaling to larger architectures.

In this work, we show that MIM models have different types of blocks: those that mainly improve the pre-training objective and others that are responsible for abstraction within the MIM encoder. The origin of this behavior can be traced back to the fact that MIM architectures usually comprise a large ViT encoder together with a *very* light-weight decoder. For larger models, the light-weight decoder reaches a point, where it cannot further improve the pre-training objective on its own and passes part of the reconstruction task back to the last encoder blocks. Consequently, the feature quality for downstream tasks of the later blocks degrades, and, somewhat unusual, the representation quality peaks in the middle blocks of the encoder.

Based on these insights, we introduce MIM-Refiner, a simple – yet highly effective – sequential refinement approach tailored to MIM models. MIM-Refiner applies an ensemble of ID heads that enforce semantic clusters via an ID objective. Most importantly, those ID heads are attached to intermediate blocks of the encoder including those that exhibit peak representation quality, instead of only a single ID head attached to the last block.

Experimentally, we show that within few epochs, MIM-Refiner refines the features of a MIM model to (i) incorporate the beneficial properties of ID objectives (ii) preserves the advantages of the MIM model (iii) exploits the synergies of both methods to improve upon each individual pre-training objective, advancing the state-of-the-art across various benchmarks, see Figure 2. Extensive evaluations show the potential of MIM-Refiner for training large-scale vision foundation models.

Our contributions can be summarized as follows:

1. We show via a detailed analysis that MIM models have different types of blocks: those that mainly improve the pre-training objective and others that are responsible for abstraction.

2. We introduce MIM-Refiner, a sequential approach to refine the representation of a pre-trained MIM model to form semantic clusters via an ID objective. Motivated by the findings in (1), MIM-Refiner is designed to exploit the intermediate representations via an ensemble of ID heads attached to multiple encoder blocks.

3. We experimentally show the effectiveness and generality of MIM-Refiner by refining a multitude of MIM models of various scales, which achieve new state-of-the-art results in a broad range of downstream tasks.

## 2 BLOCK TYPES IN MASKED IMAGE MODELING

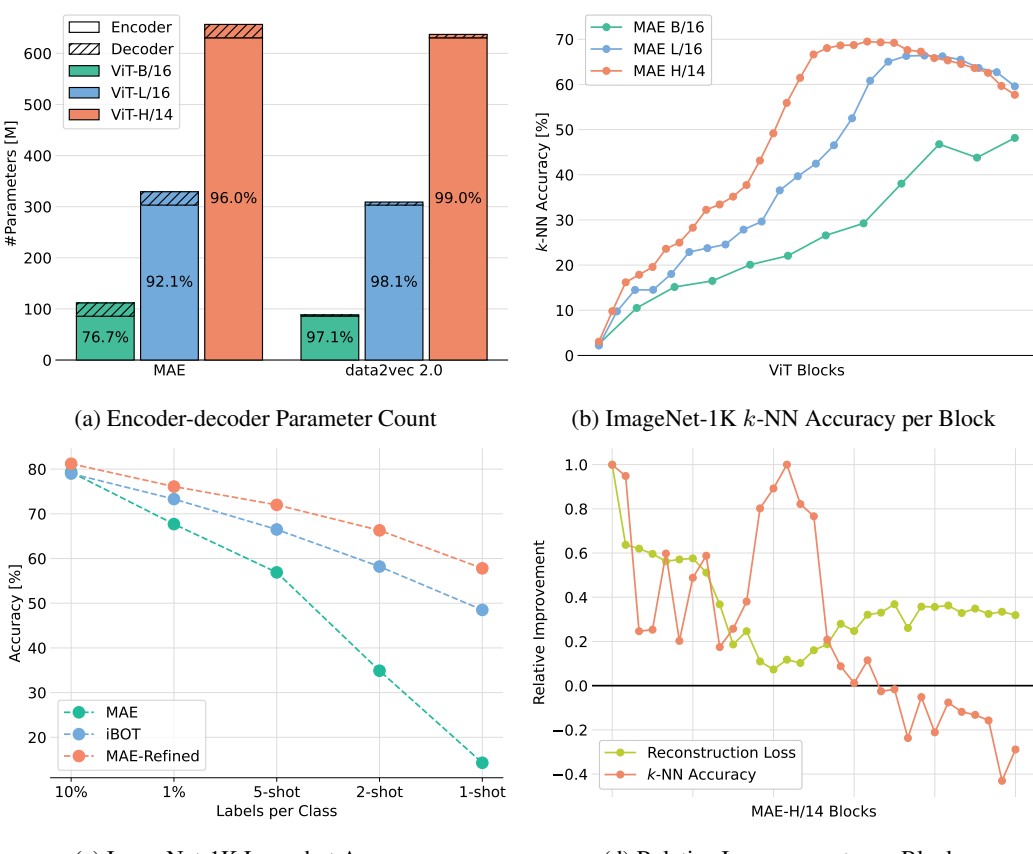

(a) Encoder-decoder Parameter Count

(b) ImageNet-1K $k$-NN Accuracy per Block

(c) ImageNet-1K Low-shot Accuracy.

(d) Relative Improvements per Block.

Figure 3: Our analysis reveals different block types in MIM models. **(a)** MIM models are asymmetrically designed where the encoder has most of the parameters, as indicated by the percentages in the bars. **(b)** The representation quality of MAE encoders (measured by $k$-NN accuracy) peaks in the middle blocks before degrading when the later blocks take over parts of the decoder's task. Various MIM models and also a semantic segmentation task follow this pattern (see Appendix B.4) **(c)** The decline in representation quality in later blocks primarily contributes to the degradation in downstream performance. ID methods (represented by iBOT) and our MAE-Refined do not suffer from this issue. **(d)** Correlation of the relative improvement of reconstruction loss and $k$-NN accuracy per block. The relative improvement is the difference between subsequent blocks divided by the maximum difference over all blocks. Figure (b) and Figure 9 in the appendix show the raw $k$-NN accuracies and reconstruction losses from which the relative improvement is obtained. Middle blocks form abstract representations (large improvements in the $k$-NN accuracy, almost no improvement in reconstruction loss), later blocks take over parts of the reconstruction task (decrease in the $k$-NN accuracy, large improvement in the reconstruction loss). Additional details can be found in Appendix D.2.

We start with an analysis, which – to the best of our knowledge – is the first that clearly reveals the different block types in MIM models. Motivated by these findings, we introduce a targeted refinement process of blocks that harm downstream performance in Section 3.

**Different blocks due to asymmetric encoder-decoder design.** MIM models, such as MAE (He et al., 2022) and data2vec 2.0 (Baevski et al., 2023) enable an efficient pre-training of large models. In terms of architecture, the encoder and decoder are intentionally designed asymmetrically. The encoder, on the one hand, is a large ViT that discards 75% of the input patches through masking to drastically reduce training costs. The decoder, on the other hand, operates on the full sequence length – by concatenating mask tokens to the encoded visible patches – and, thus, is typically *very*

lightweight to compensate for the increased number of tokens (Figure 3a). As models increase in size, the decoder eventually reaches a point where it cannot further improve the pre-training objective on its own. Consequently, it begins to delegate a portion of the reconstruction task back to the last encoder blocks. This transfer adversely affects the feature quality for downstream tasks associated with those blocks (Figure 3b), especially when only few labels are available (Figure 3c). We observe this phenomenon by correlating the relative improvement in the reconstruction loss vs the $k$-NN accuracy (Figure 3d). Roughly speaking, the blocks of the encoder operate in three different regimes:

1. In early ViT blocks, general purpose features are learned, which improve the reconstruction loss and the $k$-NN accuracy simultaneously.

2. In middle ViT blocks, abstractions are formed. The reconstruction loss improves only slightly, while the $k$-NN accuracy improves drastically.

3. In late ViT blocks, features are prepared for the reconstruction task. The reconstruction loss improves at a faster rate, while the $k$-NN accuracy decreases.

**What to learn from these findings?** Naïvely using the last encoder block features uses features suited for reconstruction, not for downstream tasks. If lots of labels are available, this can be compensated by fine-tuning the last encoder blocks on the downstream task. However, if not enough labels are available, or the last encoder blocks are not fine-tuned, downstream performance suffers.

One would think that simply using a larger decoder solves these problems. However, there are multiple problems with this solution. (i) The decoder commonly operates on the full sequence length, making it costly to increase its size. (ii) Scaling the decoder can decrease performance as shown, for example, in MAE (He et al., 2022) (Table 1). (iii) Models that can use a deeper decoder (such as CrossMAE (Fu et al., 2024)) also show degrading representation quality in later blocks (see Appendix Figure 7d). Instead of changing established MIM pre-training procedures, we ask the question: can we leverage and further improve the strong intermediate representations of MIM models?

## 3 METHOD

We propose MIM-Refiner, a novel approach aimed at improving downstream performance by refining the later blocks of a pre-trained MIM model. MIM-Refiner leverages the abstract intermediate representations with an ensemble of Instance Discrimination (ID) heads, which are attached to multiple blocks towards the end of the encoder, as visualized on the left side of Figure 4. The resulting experimental improvements in various downstream tasks are discussed in Section 4.

Inspired by NN based contrastive learning (Dwibedi et al., 2021; Azabou et al., 2021), we propose Nearest Neighbor Alignment (NNA). NN contrastive objectives introduce an inter-sample correlation by retrieving NNs of samples in a batch and subsequently applying an objective between the samples and their NNs. In practice, the NN retrieval is typically done by tracking samples from previous iterations in a first-in-first-out queue (Dwibedi et al., 2021; He et al., 2020). Therefore, the samples in the queue are from a previous model state which creates a tradeoff between the benefit of the NN-swap and the worse signal from the out-of-sync samples. We argue that the NN-swap does not offer a benefit for negative samples, since they are already different images, and instead, degrades the signal from the contrastive objective. We therefore propose to use the NN only for the alignment of positive samples, as visualized on the right side of Figure 4. Omitting the NN-swap for the negatives stabilizes training for large models and slightly improves performance (see Table 1). NNA is a variant of NNCLR and we visualize their difference in Figure 13.

Formally, given a batch of features $\mathcal{Z} = \{z_1, \ldots, z_N\}$ and a queue $\mathcal{Q}$ the NNA objective is:

$$\mathcal{L}_i^{\text{NNA}} = -\log \frac{\exp(\text{NN}(z_i, \mathcal{Q}) \cdot z_i / \tau)}{\exp(\text{NN}(z_i, \mathcal{Q}) \cdot z_i / \tau) + \sum_{j=1}^{N} \exp(\text{SG}(z_j) \cdot z_i / \tau)[i \neq j]} \quad (1)$$

$$\text{NN}(z_i, \mathcal{Q}) = \underset{q \in \mathcal{Q}}{\arg\max}(z_i \cdot q) \quad (2)$$

where $z_i$ is the anchor, $\text{NN}(z_i, \mathcal{Q})$ is the positive, $z_j$ are the negatives, $\tau$ is the temperature, and SG is the stop-gradient operation. $[i \neq j]$ denotes the Iverson bracket that evaluates to 1 if $i \neq j$ and to 0 if $i = j$. All vectors are assumed to be normalized to length 1.

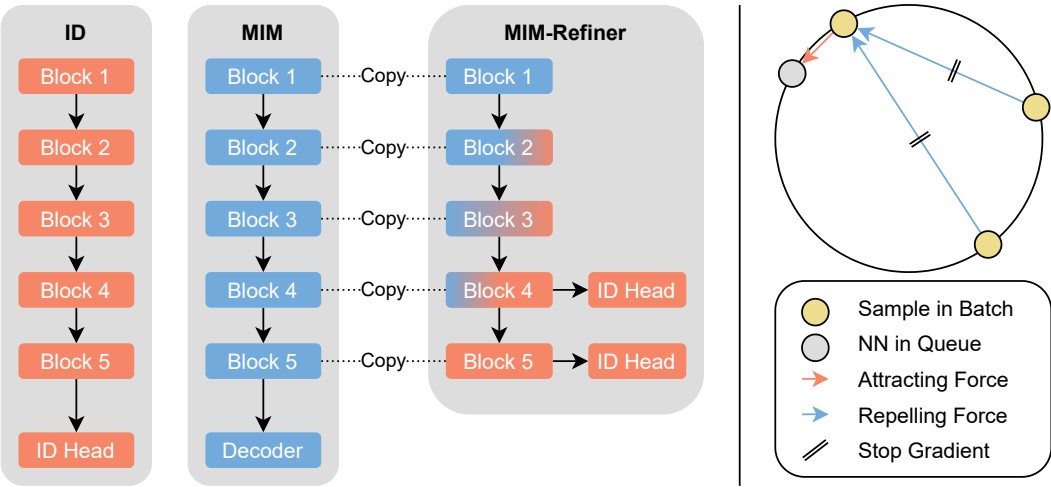

Figure 4: **Left:** Comparison of different pre-training schemes. ID uses a single ID head, whereas MIM models use a light-weight decoder to train an encoder. MIM-Refiner attaches multiple ID heads to the later third of the blocks of a pre-trained MIM encoder. **Right:** Nearest Neighbor Alignment (NNA). An anchor sample is attracted by its NN and simultaneously repelled from other samples in the batch. The NN is retrieved from a first-in-first-out queue of samples from previous batches.

## 4 EXPERIMENTS

We refine a series of MIM models, namely MAE (He et al., 2022), data2vec 2.0 (Baevski et al., 2023) (abbreviated as D2V2), dBOT (Liu et al., 2022) and CrossMAE (Fu et al., 2024). These models were pre-trained on ImageNet-1K (Deng et al., 2009), which we also use for the refinement. Models are refined for 20/30 epochs using a peak learning rate of 4e-4 with a layer-wise decay of 0.65. ID heads are attached after each block in the last third. Further implementation details are listed in Appendix C. For visual clarity, we compare our best model against previous state-of-the-art models and show the full result tables in the Appendix. We evaluate our models in classification (low- and many-shot), feature evaluation, clustering, transfer learning and semantic segmentation.

### 4.1 ABLATION STUDY

Table 1: Ablation study by refining data2vec 2.0 (Baevski et al., 2023) models. Default settings

(a) **Head Count L/16**

| #Heads | $k$-NN |
|--------|--------|
| 1 | 80.2 |
| 2 | 80.2 |
| 4 | 80.9 |
| 8 | **81.0** |
| 12 | **81.0** |

(b) **Head Count H/14**

| #Heads | $k$-NN |
|--------|--------|
| 1 | 80.7 |
| 2 | 81.1 |
| 4 | 81.7 |
| 8 | 82.1 |
| 12 | **82.3** |

(c) **Head Schedule L/16**

| Schedule | $k$-NN |
|----------|--------|
| Constant | **81.0** |
| Uniform Decay | 80.9 |
| Staggered Decay | **81.0** |
| Staggered Step | 80.9 |
| One Hot | 80.7 |

(d) **NN-swap L/16**

| Swap Neg. | $k$-NN |
|-----------|--------|
| ✗ | **81.0** |
| ✓ | 80.9 |

(e) **NN-swap H/14**

| Swap Neg. | $k$-NN |
|-----------|--------|
| ✗ | **82.3** |
| ✓ | 80.7 |

(f) **Augmentations L/16**

| Color/blur | $k$-NN | 1-shot |
|------------|--------|--------|
| ✓ | **81.0** | 61.7 |
| ✗ | 80.5 | **63.4** |

(a) Adding **multiple heads** at intermediate features improves $k$-NN accuracy by 0.8%. The best settings include the blocks with the best representation quality (see Figure 3b). Adding additional heads before the best blocks does not improve performance while increasing training costs. (b) The benefit **doubles to 1.6% for a ViT-H** since deeper models degrade more (see Figure 3b). Additionally,

we investigate where to attach the ID heads in Appendix B.2 where we find that simply attaching ID head to the last 8 (or 12 for ViT-H) blocks to perform best across model scales.

(c) **Scheduling the loss weight** of each intermediate head is not necessary, i.e. simply summing all losses is sufficient to achieve good performances. "Uniform Decay" decays the loss weight of all intermediate heads during training. "Staggered Decay" starts by decaying the first head and gradually starts to decay more and more heads as training progresses. "Staggered Step" disables the first head after some time followed by gradually disabling more and more heads. "One Hot" trains the encoder with only one intermediate head at a time where training starts with the earliest intermediate head and gradually iterates through the heads. Details to the loss schedules are in Appendix D.14.

(d) Using the **NN swap** only for aligning the positive with the anchor gives a small but consistent improvement on smaller models. (e) The improved signal quality is **cruicial to avoid training instabilities** in larger models. These instabilities manifest in a sudden representation quality drop mid-training leading to a much worse final model.

(f) Relying only on the data-driven **augmentation** of the NN-swap by omitting color/blur augmentations, is beneficial for certain downstream tasks such as extreme low-shot classification (Lehner et al., 2024). We show more results without color/blur augmentations in Appendix B.5.

## 4.2 LOW-SHOT AND FEATURE EVALUATIONS

We evaluate the ability of our models to perform low-shot classification in Table 2. Additionally, linear probing and $k$-NN accuracy are reported which are computed from the features of the frozen encoder. These metrics are typically correlated to low-shot performance as it indicates that the representation is already linear separable which eases drawing decision boundaries given only few labels. For linear probing, we use the protocol of DINOv2 (Oquab et al., 2023) which includes using features of the last four blocks in its hyperparameter grid. Therefore, linear probing evaluates the features at the end of the encoder, while $k$-NN accuracy evaluates only the features of the last block.

MIM-Refiner drastically improves upon MIM models and other SSL models. In the 1-shot settings D2V2-Refined-H sets a new state-of-the-art of 64.2%, outperforming the 63.6% of MAWS-6.5B (Singh et al., 2023) which is pre-trained on a private dataset with 2000x the size of ImageNet-1K.

Table 2: Low-shot and feature evaluations of recent SSL models on ImageNet-1K. MIM-Refiner significantly improves upon MIM models and previous state-of-the-art SSL models. Appendix Table 10 extends this comparison to more methods and to models that were trained on more data (such as DINOv2) where MIM-Refiner outperforms DINOv2-g in some benchmarks.

| ViT | Method | Low-shot Evaluation | | | | | Feature Eval | |
|-----|--------|--------|--------|--------|------|------|-------|-------|
| | | 1-shot | 2-shot | 5-shot | 1% | 10% | Probe | $k$-NN |
| L/16 | MAE | 14.3 | 34.9 | 56.9 | 67.7 | 79.3 | 77.5 | 60.6 |
| | D2V2 | 24.1 | 58.8 | 72.1 | 75.1 | 81.5 | 78.2 | 51.8 |
| | iBOT | 48.5 | 58.2 | 66.5 | 73.3 | 79.0 | 81.1 | 78.0 |
| | Mugs | 52.9 | 62.3 | 69.4 | 76.2 | 80.3 | 82.1 | 80.4 |
| | MAE-Refined | 57.8 | 66.3 | 72.0 | 76.1 | 81.2 | 82.8 | **81.5** |
| | D2V2-Refined | **61.7** | **69.6** | **73.9** | **78.1** | **82.1** | **83.5** | 81.0 |
| H/14 | MAE | 7.2 | 14.1 | 40.2 | 72.8 | 81.2 | 78.2 | 58.1 |
| | D2V2 | 21.6 | 60.8 | 74.2 | 77.6 | 83.3 | 80.4 | 48.0 |
| | MAE-CT | 49.4 | 59.6 | 67.4 | 74.4 | 81.7 | 82.3 | 79.1 |
| | MAE-Refined | 59.5 | 68.5 | 73.8 | 77.4 | 82.1 | 83.7 | **82.3** |
| | D2V2-Refined | **64.2** | **71.3** | **75.5** | **78.1** | **83.5** | **84.7** | **82.3** |
| 2B/14 | MAE | 17.8 | 29.1 | 62.9 | 73.6 | 82.0 | 79.7 | 67.1 |
| | MAE-Refined | **58.2** | **68.6** | **74.8** | **78.7** | **82.5** | **84.5** | **83.2** |

## 4.3 CLUSTER EVALUATIONS

We compare the cluster performance of MIM-Refiner against recent SSL models in Table 3. We apply mini-batch $k$-means (Sculley, 2010) 100 times to the validation set of ImageNet-1K and select

Table 3: $k$-means cluster performance and class separation on ImageNet of recent SSL models. MIM-Refiner drastically improves performance of MIM-models and even outperforms DINOv2-g which has 2x more parameters and is trained on on 100x more data.

| ViT | Method | Cluster Performance | | | | Class Separation | |
|---|---|---|---|---|---|---|---|
| | | ACC | NMI | AMI | ARI | SIL ($\uparrow$) | DBS ($\downarrow$) |
| L/16 | D2V2 | 10.5 | 45.1 | 19.5 | 2.5 | -9.1 | 6.4 |
| | iBOT | 52.2 | 80.5 | 67.0 | 33.4 | 13.3 | 3.5 |
| | Mugs | 54.3 | 78.6 | 65.5 | 22.4 | 14.9 | 3.3 |
| | D2V2-Refined | **67.4** | **86.3** | **76.2** | **40.5** | **37.1** | **2.2** |
| H/14 | D2V2 | 9.9 | 45.8 | 18.0 | 2.6 | -10.8 | 6.5 |
| | D2V2-Refined | **67.3** | **87.2** | **77.9** | **42.2** | **34.5** | **2.3** |
| H/16 | MAE-CT | 58.0 | 81.8 | 69.3 | 36.8 | - | - |
| g/14 | DINOv2 (LVD-142M) | 47.7 | 76.3 | 63.6 | 5.1 | 30.4 | 2.8 |

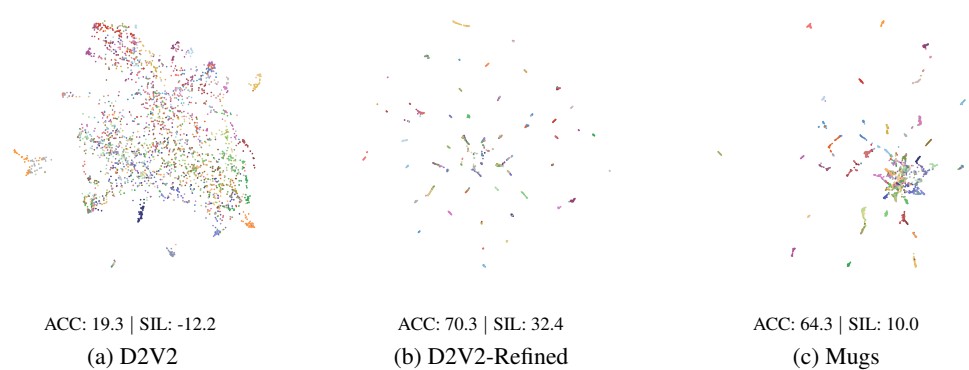

| ACC: 19.3 \| SIL: -12.2 | ACC: 70.3 \| SIL: 32.4 | ACC: 64.3 \| SIL: 10.0 |
|---|---|---|
| (a) D2V2 | (b) D2V2-Refined | (c) Mugs |

Figure 5: UMAP (McInnes et al., 2018) plots of ViT-L embeddings using all 53 food related classes of ImageNet. The corresponding $k$-means cluster accuracy (ACC) and class separation measured in silhouette score (SIL) is shown below each plot. The clustering after refinement (b) is visually more condensed and better separated with corresponding improvements in ACC and SIL than before refinement (a). Mugs **(c)** does not separate the clusters that well, as shown by the merged clusters in the middle and the lower SIL score. The colors show the 53 ground truth food classes.

the run with the lowest $k$-means loss for comparison (average performance is reported in Appendix Table 12). We report commonly used metrics for measuring *Cluster Performance* w.r.t. the ground truth clustering: Cluster Accuracy (ACC) (Yang et al., 2010), Normalized Mutual Information (NMI) (Kvalseth, 1987), Adjusted Mutual Information (AMI) (Nguyen et al., 2009), Adjusted Rand Index (ARI) (Hubert & Arabie, 1985), where higher values indicate a better match of the found clustering with the ground truth labels. Further, we measure the *Class Separation* in the embedded space using the Davies-Bouldin score (DBS) (Davies & Bouldin, 1979) and silhouette score (SIL) (Rousseeuw, 1987) w.r.t. the ground truth classes. The DBS measures the separation and compactness of classes, where lower values are better and zero is the minimum. The SIL ranges from -100 to 100, where higher values are better. Negative SILs relate to mismatches between classes and embedding, where scores close to zero indicate a potential overlap of classes.

Table 3 shows that MIM-Refiner greatly improves various clustering metrics. The reached ACC of 67.4% outperforms the current state-of-the-art of 61.6% reached by TEMI MSN (Adaloglou et al., 2023). Figure 5 illustrate this drastic increase in cluster performance and class separation visually. Additionally, we evaluate combining multiple models using the TURTLE (Gadetsky et al., 2024) framework in Appendix Table 13.

Table 4: Transferring MIM-Refiner models to other datasets. "VTAB-6" reports the average accuracy over six datasets from the VTAB benchmark (Zhai et al., 2019) and "VTAB-1K" is the average over all 19 datasets of the VTAB-1K benchmark. ADE20K reports the mean intersection over union (mIoU) of a semantic segmentation probe. MIM-Refiner learns general features that can easily be transferred to various datasets and tasks. Table 15 confirms this finding on additional models and Appendix B.11 shows individual VTAB performances for VTAB-6 probing and VTAB-1K fine-tuning.

| | | iNat18 Fine-tuning | | | Linear Probe | | | VTAB-1K |
|---|---|---|---|---|---|---|---|---|
| ViT | Method | 1-shot | 5-shot | 10-shot | iNat18 | VTAB-6 | ADE20K | Fine-tuning |
| L/16 | MAE | 7.1 | 51.6 | 68.9 | 42.8 | 83.0 | 33.6 | 73.7 |
| | iBOT | 15.8 | 51.5 | 65.5 | 56.0 | 87.6 | 35.6 | 70.8 |
| | Mugs | **19.5** | 53.2 | 66.9 | **61.5** | 87.9 | 34.8 | 68.0 |
| | MAE-Ref. | 19.0 | **58.0** | **71.7** | 60.6 | **88.5** | **37.3** | **75.2** |
| H/14 | MAE | 6.5 | 53.3 | 71.7 | 43.0 | 83.2 | 35.5 | 72.7 |
| | MAE-CT | 16.5 | 60.1 | 74.7 | 62.8 | 88.8 | 37.6 | 75.7 |
| | MAE-Ref. | **20.9** | **62.4** | **75.4** | **64.6** | **89.3** | **39.4** | **75.9** |
| 2B/14 | MAE | 10.0 | 53.7 | 72.2 | 51.0 | 85.4 | 37.3 | 74.1 |
| | MAE-Ref. | **22.5** | **63.5** | **76.5** | **69.6** | **89.8** | **40.3** | **75.6** |

## 4.4 TRANSFER LEARNING EVALUATIONS

We investigate generalization of pre-trained MIM-Refiner models to other datasets in Table 4, which shows the benefits of MIM-Refiner models when transferring a pre-trained representation. We consider a variety of classification downstream tasks and a semantic segmentation task. MAE-Refined consistently improves over MAE and state-of-the-art SSL methods. Additionally MIM-Refiner further improves when scaling up to a 2B parameter model.

## 4.5 FINE-TUNING WITH LARGE AMOUNTS OF LABELS

MIM models typically outperform ID methods when given enough labels to fine-tune the model. As our refinement process employs an ID objective, we investigate whether MIM-Refiner involuntarily degrades performance given an abundance of labels. To this end, we fine-tune MIM models and their refined version on ImageNet-1K (Deng et al., 2009), iNat18 (Horn et al., 2018) and ADE20K (Zhou et al., 2019) using a linear classification or UperNet (Xiao et al., 2018) segmentation head. Table 5 shows a small but consistent improvement of MIM-Refiner models.

Table 5: Full fine-tuning using 100% of the labels. MIM-Refiner consistently improves performance slightly even though ID methods perform worse than MIM models on this benchmark. Appendix Table 18 confirms this pattern on additional MIM models. "Robustness" shows the average performance of the trained IN-1K classifier on robustness datasets (individual results in Table 19). ID and JEPA (Assran et al., 2023) models (shown in gray) are not competitive with MIM(-Refiner) models.

| | ViT-L/16 | | | | ViT-H/14 | | |
|---|---|---|---|---|---|---|---|
| Model | IN-1K | Robustness | iNat18 | ADE20K | IN-1K | Robustness | iNat18 |
| D2V2 | 86.6 | 60.2 | 81.0 | **54.4** | 86.6 | 63.2 | 79.6 |
| D2V2-Ref. | **86.7** | **60.5** | **81.6** | **54.4** | **86.8** | **64.1** | **79.8** |
| dBOT | 85.8 | **55.3** | 81.9 | 53.1 | **87.1** | 63.7 | 84.1 |
| dBOT-Ref. | **85.9** | **55.3** | **82.1** | **53.3** | **87.1** | **64.0** | **84.5** |
| iBOT | 84.8 | 47.7 | 76.9 | 51.1 | - | - | - |
| Mugs | 85.2 | 46.4 | 76.9 | 50.2 | - | - | - |
| I-JEPA | - | - | - | - | 84.9 | 50.9 | 75.9 |

While improvements in fine-tuning with large amounts of labels might seem marginal when compared to the gains of MIM-Refiner on other benchmarks, it is important to consider that these benchmarks are extremely competitive and advancements thereon are made in small increments. Gains of one

percentage point or larger are unrealistic when keeping data and model size constant. Additionally, MIM-Refiner is designed to be compute efficient, requiring only a couple of epochs of training, whereas training from scratch requires multiple hundred epochs. The fact that MIM-Refiner still improves fine-tuning with large amounts of labels slightly shows that our method can efficiently learn strong semantic representations, even improving upon MIM models in their "strong suit" (fine-tuning with large amounts of labels).

### 4.6 DISCUSSION & FUTURE WORK

We experimentally show that MIM-Refiner efficiently combines the advantages of MIM and ID methods without suffering from their respective disadvantages. When comparing to the current state-of-the-art models in ID (iBOT, Mugs), MIM-Refiner consistently outperforms them across a variety of benchmarks, most of the time by large margins. Additionally, the efficiency of MIM-Refiner allows us to scale up model size beyond the largest ID models, without requiring unreasonable amounts of compute. To put it into perspective, the currently largest ID model trained on ImageNet-1K are ViT-L models (300M parameters) whereas MIM-Refiner can effortlessly scale up to 2B parameter models. Even DINOv2 Oquab et al. (2023), a vision foundation model trained on a private high-quality dataset with 142M images, trains only a 1B parameter model. When comparing against the current state-of-the-art MIM models, MIM-Refiner drastically improves performance in few-shot settings while also showing slight improvements in many-shot settings. Overall, we have not found a single setting where refining MIM models would be undesirable.

MIM-Refiner shows strong scaling behavior, however we are only able to show results on ImageNet-1K due to resource constraints. When scaling up to ImageNet-21K, we would first need to pre-train large-scale MIM models ourselves, including tuning hyperparameters (learning rate, training duration, . . . ). This requires a lot of compute, which is not within our current compute budget. Note that for ImageNet-1K, we simply download the pre-trained checkpoints and only need compute for the refinement process. Additionally, going beyond ImageNet-21K to web-scale datasets is also heavily restricted by the lack of publicly available high-quality datasets for ID training (such as LVD-142M). However, MIM and ID models have been shown to scale well to larger datasets (Singh et al., 2023; Oquab et al., 2023), which suggests that also MIM-Refiner would scale well to larger datasets.

## 5 RELATED WORK

### 5.1 PRE-TRAINING IN COMPUTER VISION

Following the success of generative pre-training of transformers (Vaswani et al., 2017) in language modeling (Devlin et al., 2019; Radford et al., 2018), similar directions were explored in computer vision (Dosovitskiy et al., 2021; Xie et al., 2022; Wei et al., 2022; Chen et al., 2020a). With the introduction of Vision Transformers (Dosovitskiy et al., 2021), large Masked Image Modeling (MIM) models could be efficiently trained (He et al., 2022; Baevski et al., 2023; Singh et al., 2023) by using the ability of transformers to effortlessly process sparse input by dropping masked patch tokens in the input and subsequently reconstructing the masked parts. In order to optimize the MIM pre-training objective, models have to infer the missing regions by efficiently encoding foreground and background alike which leads to a rich representation.

Building on rich features learned by MIM models has been explored in various ways. MAWS (Singh et al., 2023) first pre-trains an MAE, followed by weakly supervised training on a billion-scale dataset using billion-scale models. SemiViT (Cai et al., 2022) uses a pre-trained MAE as a starting point for semi-supervised learning. Segment Anything (Kirillov et al., 2023) use MAE as basis for a segmentation foundation model. MIM-Refiner also builds on the rich features of MIM models and refines them with a ID objective to ease adaption to downstream tasks.

Instance Discrimination (ID) is another branch of self-supervised learning that uses augmentations to create multiple views of the same sample where the task is then to find matching pairs within the views from all samples within a batch (Chen et al., 2020c; He et al., 2020; Dosovitskiy et al., 2016; Wu et al., 2018) or align features of one view with the features from another (Grill et al., 2020; Caron et al., 2021). We use the terminology that views from the same sample are "positive pairs" and views of different samples are "negative pairs". When describing a single view of a sample, it is called the "anchor" to which all other views in a batch are either "positives" or "negatives".

NN-based ID (Dwibedi et al., 2021; Azabou et al., 2021) extends this setting to use NNs in various ways to create views or otherwise augment samples during training. MIM-Refiner introduces Nearest Neighbor Alignment, which is a modification of previous NN-based ID methods to use the NN only for the alignment part, i.e. pulling the anchor closer to the NN of its positives while pushing the anchor away from its negatives.

## 5.2 Combining MIM and ID

Adding a MIM to ID methods has emerged as a powerful pre-training scheme. However, in contrast to ID models, the MIM objective in the end-to-end training either uses significantly lower masking ratios with mask tokens getting processed by the encoder (Zhou et al., 2021; Oquab et al., 2023), or requires a target encoder to encode the unmasked image (Huang et al., 2022b; Assran et al., 2022). Both *drastically* increase the computational requirements as the encoder operates on the full sequence length. Consequently, these models either require copious amounts of compute to train (Oquab et al., 2023) or limit themselves to relatively small models (Zhou et al., 2021; Huang et al., 2022b). Contrary, MIM models have shown success and scalability to large models with comparatively little compute (He et al., 2022; Baevski et al., 2023; Singh et al., 2023). MIM-Refiner can build on these models which allows us to scale to large model sizes without a large amount of compute. Our largest model, MAE-Refined-2B contains approximately twice the parameters of the currently largest uni-modal contrastive model DINOv2-g (Oquab et al., 2023) and can be trained on two orders of magnitude less data.

Attempts to preserve the efficiency while training MIM and ID objectives end-to-end have been less successful (Lehner et al., 2024; Jiang et al., 2023), where both works came to the conclusion that sequential training (MIM → ID) circumvents various problem with end-to-end training. First, a powerful encoder is trained solely with a MIM objective. Second, the encoder is trained with a ID objective while preserving the rich features in early blocks with a layer-wise learning rate decay (Clark et al., 2020) in lower blocks and either constraining changes in early blocks (Jiang et al., 2023) or completely freezing them (Lehner et al., 2024).

MIM-Refiner is also a sequential MIM → ID method. In contrast to our work, previous works start from the representation after the last MIM encoder block and are therefore highly reliant on a good representation thereof. This can be seen for example on MAE-CT (Lehner et al., 2024) where their ViT-H/14 model is worse than their ViT-H/16 model, despite using 30% more FLOPS. Additionally, previous works (Lehner et al., 2024; Jiang et al., 2023) omit current go-to techniques such as multi-crop augmentation (Caron et al., 2020) which has been shown to improve performance (Caron et al., 2021; Zhou et al., 2021; 2022).

Training models with additional losses from intermediate layers dates back to the early deep learning days (Lee et al., 2015; Szegedy et al., 2015) where these auxiliary losses were used to alleviate optimization issues in lower layers. MIM-Refiner relates to deep supervision in the sense that we use multiple ID heads attached at intermediate layers where each head produces a loss for training.

## 6 Conclusion

We introduce MIM-Refiner, a procedure to refine pre-trained MIM models. Motivated by the insights that the representation quality of MIM models peaks in the middle of the encoder, we employ an ensemble of instance discrimination heads attached at multiple blocks towards the end of the encoder, including the blocks where representation quality peaks, to improve upon the best existing representation. We train this ensemble for a short duration to improve the representation for downstream tasks such as classification, clustering or semantic segmentation.

Our refined MIM models learn strong features from ImageNet-1K alone that can be readily used for downstream tasks without fine-tuning the model but also improve performance when the model is fine-tuned, particularly in few-shot settings. Our models outperform competitors that were also trained on ImageNet-1K and sometimes also ones that were trained on more data or use larger models.

Extensive evaluations show the potential of MIM-Refiner for training large-scale vision foundation models that provide general-purpose off-the-shelf features for a broad range of downstream tasks.

## REPRODUCIBILITY

The full codebase used for all experiments in this paper together with the exact hyperparameter configurations that were used for each experiment and pre-trained models can be found in our github repository: https://github.com/ml-jku/MIM-Refiner.

We provide the hyperparameters for training and evaluation pipelines in Appendix Section D together with additional implementation details in Appendix Section C.

## ACKNOWLEDGEMENTS

We thank Johannes Lehner for helpful discussions and suggestions.

We acknowledge EuroHPC Joint Undertaking for awarding us access to Karolina at IT4Innovations, Czech Republic, MeluXina at LuxProvide, Luxembourg, LUMI at CSC, Finland and Leonardo at CINECA, Italy. We acknowledge access to LEONARDO at CINECA, Italy, via an AURELEO (Austrian Users at LEONARDO supercomputer) project.

The ELLIS Unit Linz, the LIT AI Lab, the Institute for Machine Learning, are supported by the Federal State Upper Austria. We thank the projects Medical Cognitive Computing Center (MC3), INCONTROL-RL (FFG-881064), PRIMAL (FFG-873979), S3AI (FFG-872172), EPILEPSIA (FFG-892171), AIRI FG 9-N (FWF-36284, FWF-36235), AI4GreenHeatingGrids (FFG- 899943), IN-TEGRATE (FFG-892418), ELISE (H2020-ICT-2019-3 ID: 951847), Stars4Waters (HORIZON-CL6-2021-CLIMATE-01-01). We thank Audi.JKU Deep Learning Center, TGW LOGISTICS GROUP GMBH, Silicon Austria Labs (SAL), FILL Gesellschaft mbH, Anyline GmbH, Google, ZF Friedrichshafen AG, Robert Bosch GmbH, UCB Biopharma SRL, Merck Healthcare KGaA, Verbund AG, GLS (Univ. Waterloo), Software Competence Center Hagenberg GmbH, Borealis AG, TÜV Austria, Frauscher Sensonic, TRUMPF and the NVIDIA Corporation.

## ETHICS STATEMENT

Our work proposes a training methodology that can readily reuse existing models, which makes use of the expended energy for large-scale pre-training of MIM models instead of training models from scratch, which would need much more energy than refining existing models. Additionally, our models learn strong general-purpose features that can be readily used for a broad range of downstream tasks without needing to fine-tune or retrain a model. This further reduces the energy consumption and carbon emissions required for training state-of-the-art models in computer vision tasks.

The strong general-purpose features of our models provide a fundamental advancement to computer vision, inheriting their potential benefits. On the one hand, our models could be used for environmental monitoring of endangered species where little data is available. On the other hand, the strong few-shot performance of our models could lead to overconfidence in rare disease classifiers where sufficient data is indispensable and checking with a healthcare professional is necessary to avoid misdiagnoses.

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

# A    LIMITATIONS

One limitation of MIM-Refiner is that it requires batch normalization (Ioffe & Szegedy, 2015) layers in the ID head. Without them, performance decreases significantly. Similar observations have been made in other contrastive learning literature Chen et al. (2021). The batch normalization layers significantly decrease scalability to distributed hardware setups as each layer requires a synchronization of batch statistics Chen et al. (2020c).

MIM-Refiner addresses a common issue with MIM models: their typically lightweight decoder often delegates part of the reconstruction to the encoder, resulting in subpar representations for the later encoder blocks in downstream tasks. Alternatively, one could simply argue for a larger decoder. However, a larger decoder increases computational costs since the decoder typically operates on the full sequence length. Additionally, the direction of a larger decoder was explored to a certain extent in the original MAE paper (He et al., 2022), where larger decoders performed worse in fine-tuning and linear probing. Successor models such as CrossMAE perform well with a deeper decoder but still show decreasing representation quality in later layers, as shown in Appendix B.4.

During early development, we tried various ways to propagate the intermediate representation towards the end. While we found that a simple ensemble of contrastive heads attached to later blocks works very well, there might be even better ways to leverage the rich intermediate MIM representations. We explored the following variants in with little success: (i) completely deleting the last few blocks (ii) reinitializing the last few blocks while setting the weights/biases of the last projection in the attention/MLP to 0 which leads to the result of the previous block being propagated to the end via the residual connection (iii) gating the last few blocks via ReZero (Bachlechner et al., 2021).

To address the limitation of MIM-Refiner requiring batch normalization layers, we explore another variant that copies the weights of the intermediate block with the highest $k$-NN accuracy to all subsequent blocks and sets the weights/biases of the last projection in the copied attention/MLP blocks to 0. This is similar to the above mentioned approach (ii), except that the weights of the copied blocks are not random but copied from an intermediate block. We then train the model with only a single head attached at the last block. This drastically reduces the number of batch normalization layers. Table 6 shows that such an approach can yield competitive performances on smaller models, but is outperformed by MIM-Refiner on larger ones.

Table 6: Copying the peak-representation block to subsequent blocks while setting the last attention/MLP projection weights/biases to 0 can improve scalability to even larger distributed setups due to requiring less batch normalization layers. This approach ("Copy Blocks") is competitive to an ensemble of ID heads ("MIM-Refiner") on smaller models but is worse on larger models.

| $k$-NN | MAE L/16 | MAE H/14 |
|---|---|---|
| Copy Blocks | **81.5** | 81.8 |
| MIM-Refiner | **81.5** | **82.3** |

Another approach to improve scalability to larger distributed systems is to only aggregate batch statistics within a node. This avoids costly inter-node communication and instead only requires intra-node connections which is typically much faster.

As MIM-Refiner is quite computationally efficient even with the batchnorm synchronization limitation, we do not explore these approaches further.

## B EXTENDED RESULTS

### B.1 EXTENDED REPRESENTATION QUALITY COMPARISON

We compare linear probing, 1-shot classification performance and runtime against various image models (including non-public ones) in Figure 6.

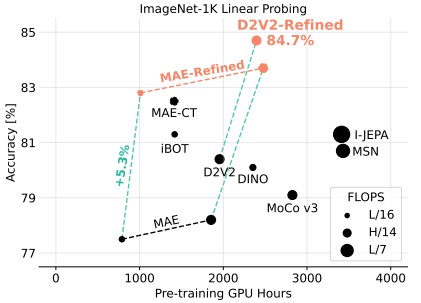 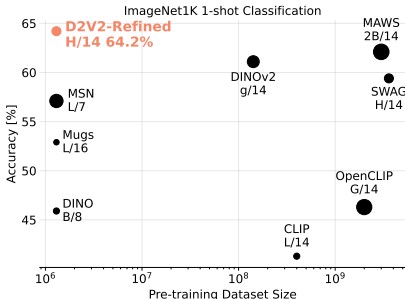

Figure 6: **Left:** Representation quality of SSL methods evaluated via linear probing. MIM-Refiner advances state-of-the-art among models pre-trained on ImageNet-1K in low- and high-compute regimes. **Right:** MIM-Refiner advances state-of-the-art in the extreme setting of 1-shot classification despite being trained on orders of magnitude less data. Size of dots corresponds to model FLOPS.

### B.2 WHERE TO ATTACH ID HEADS?

Table 7 ablates different choices of where to attach ID heads on a D2V2 pre-trained ViT-L/16 and ViT-H/14. On ViT-L/16, there is almost no difference of where to attach the ID heads in the last third of blocks. We tried to transfer this insight to ViT-H/14 where the default setting of attaching ID heads to all later blocks performs better. We therefore use the default setting of attaching ID heads to the last 8 blocks (ViT-L and ViT-2B) or the last 12 blocks (ViT-H).

Table 7: Spacing heads out more across the later ViT blocks can achieve comparable performances for ViT-L/16 but does not generalize to ViT-H/14. The default setting of attaching ID heads to all later blocks generalizes well across model scales.

| Block Indices | $k$-NN |
|---|---|
| 20,24 | 80.9 |
| 16,20,24 | **81.1** |
| 15,18,21,24 | **81.1** |
| 18,20,22,24 | 81.0 |
| 17-24 | 81.0 |

(a) ViT-L/16

| Block Indices | $k$-NN |
|---|---|
| 22,24,26,28,30,32 | 82.0 |
| 16,18,20,22,24,26,28,30,32 | 82.1 |
| 20-32 | **82.3** |

(b) ViT-H/14

### B.3 FREEZING EARLY BLOCKS

Freezing early blocks as a form of regularization to preserve MIM features (similar to related works (Lehner et al., 2024; Jiang et al., 2023)) is not necessary, as shown in Table 8. Note that we still use a layer-wise learning rate decay (Clark et al., 2020).

Table 8: Freezing early blocks is not necessary and slightly decreases performance. Ablation conducted with D2V2-L/16. We freeze the first 6 layers to refine MAE-2B to save memory/compute.

| #Frozen | $k$-NN |
|---|---|
| 0 | **81.0** |
| 6 | 80.9 |
| 12 | 80.6 |

## B.4 INTERMEDIATE REPRESENTATION ANALYSIS OF ADDITIONAL MIM METHODS

We visualize the $k$-NN accuracies of various MIM models in Figure 7 where all of them show the pattern that the representation quality of larger models degrades towards the end of the encoder.

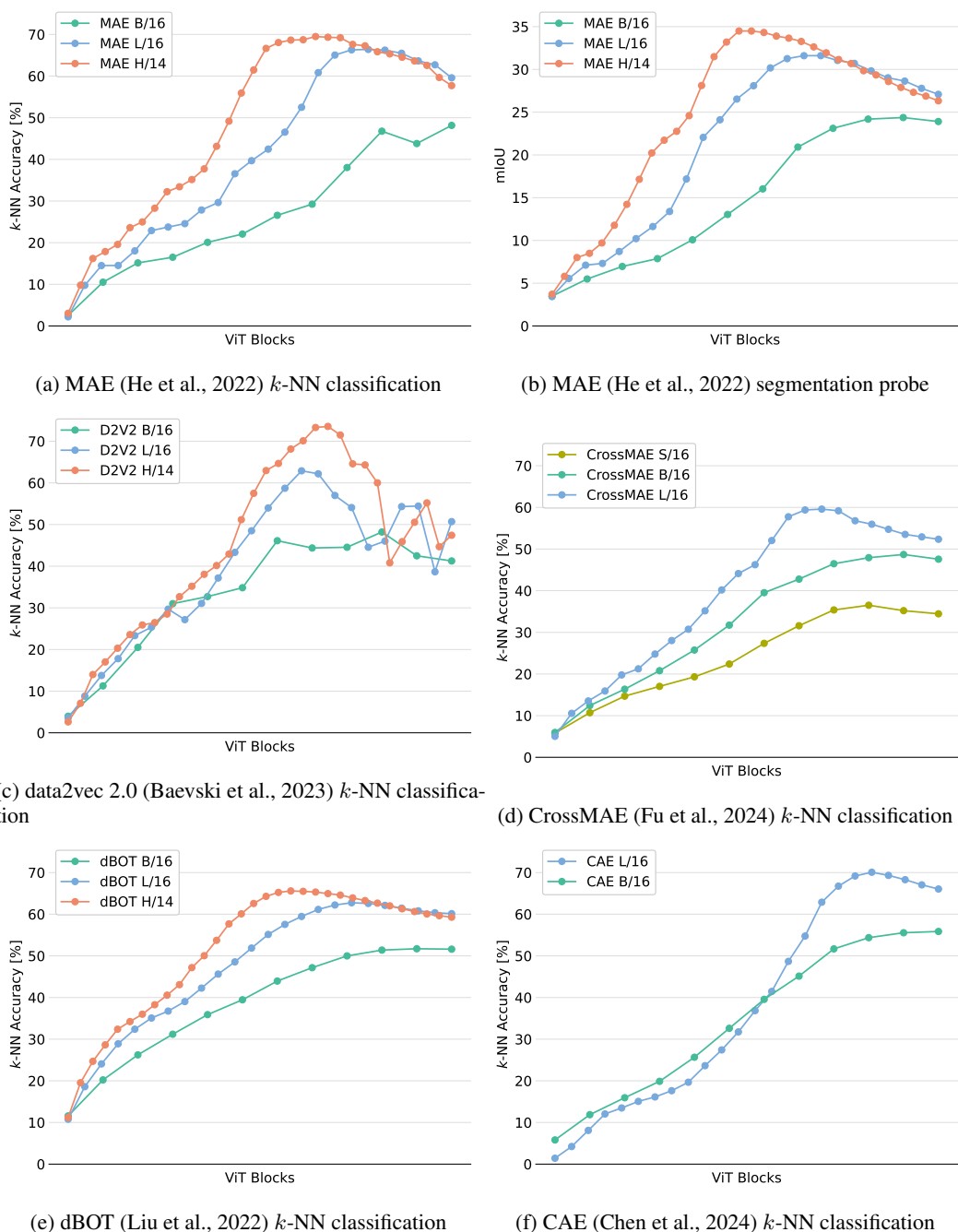

(a) MAE (He et al., 2022) $k$-NN classification

(b) MAE (He et al., 2022) segmentation probe

(c) data2vec 2.0 (Baevski et al., 2023) $k$-NN classification

(d) CrossMAE (Fu et al., 2024) $k$-NN classification

(e) dBOT (Liu et al., 2022) $k$-NN classification

(f) CAE (Chen et al., 2024) $k$-NN classification

Figure 7: Intermediate representation analysis of various MIM models. As models get bigger, the $k$-NN accuracy degrades more towards the end of the encoder, especially for L/16 and H/14 models. This pattern is consistent over various MIM models and across tasks. $k$-NN accuracy is calculated on ImageNet-1K (Deng et al., 2009). (b) shows the mIoU of linear segmentation probes on ADE20K (Zhou et al., 2019).

### B.5 LOW-SHOT CLASSIFICATION WITHOUT COLOR AUGMENTATIONS

MAE-CT (Lehner et al., 2024) showed that omitting color augmentations can lead to performance gains for low-shot classification, especially on larger models. We therefore train our ViT-H/14 models also without color augmentations. We use the same hyperparameters as with color augmentations (see Table 24) except that we disable color augmentations (i.e. we train with only crop & flip augmentations) and half the training duration. Table 9 confirms the findings of (Lehner et al., 2024) as omitting color augmentations also improves low-shot performance of MIM-Refiner.

Table 9: ImageNet-1K low-shot and feature evaluations of a D2V2-Refined-H/14 with and without color augmentations. Omitting color augmentations improves ImageNet-1K low-shot performance.

| Model | Color/blur | Low-shot Evaluation | | | | | Feature Eval | |
|---|---|---|---|---|---|---|---|---|
| | | 1-shot | 2-shot | 5-shot | 1% | 10% | Probe | $k$-NN |
| D2V2-Refined | ✗ | **64.7** | **72.0** | **75.9** | **79.1** | **83.5** | 84.1 | 82.1 |
| | ✓ | 64.2 | 71.3 | 75.5 | 78.1 | **83.5** | **84.7** | **82.3** |

### B.6 EXTENDED IMAGENET-1K LOW-SHOT AND FEATURE EVALUATIONS

Table 10: Extending Table 2 with additional MIM-Refiner models and more SSL models. The last row-group compares the best MIM-Refiner model to models with longer sequence length (MSN-L/7) and models that were pre-trained on more data. Parentheses show pre-training dataset size. MIM-Refiner consistently improves MIM models, outperforms other SSL models with the same pre-training data and even outperforms DINOv2-g/14 in some settings.

| ViT | Method | Low-shot Evaluation | | | | | Feature Eval | |
|---|---|---|---|---|---|---|---|---|
| | | 1-shot | 2-shot | 5-shot | 1% | 10% | Probe | $k$-NN |
| | CrossMAE | 16.8 | 34.0 | 52.4 | 63.2 | 77.7 | 74.4 | 53.4 |
| | MAE | 14.3 | 34.9 | 56.9 | 67.7 | 79.3 | 77.5 | 60.6 |
| | dBOT | 28.0 | 46.9 | 62.4 | 70.0 | 80.2 | 77.8 | 61.3 |
| | D2V2 | 24.1 | 58.8 | 72.1 | 75.1 | 81.5 | 78.2 | 51.8 |
| | CAE | 28.1 | 56.5 | 68.4 | 71.4 | 79.5 | 80.0 | 66.9 |
| L/16 | iBOT | 48.5 | 58.2 | 66.5 | 73.3 | 79.0 | 81.1 | 78.0 |
| | Mugs | 52.9 | 62.3 | 69.4 | 76.2 | 80.3 | 82.1 | 80.4 |
| | CrossMAE-Refined | 50.3 | 60.9 | 68.2 | 71.7 | 79.3 | 81.8 | 79.9 |
| | MAE-Refined | 57.8 | 66.3 | 72.0 | 76.1 | 81.2 | 82.8 | **81.5** |
| | dBOT-Refined | 57.4 | 66.0 | 71.7 | 76.6 | 81.6 | 83.3 | 81.3 |
| | D2V2-Refined | **61.7** | **69.6** | **73.9** | **78.1** | **82.1** | **83.5** | 81.0 |
| | MAE | 7.2 | 14.1 | 40.2 | 72.8 | 81.2 | 78.2 | 58.1 |
| | dBOT | 23.9 | 45.0 | 63.0 | 73.0 | 82.1 | 79.0 | 60.0 |
| | D2V2 | 21.6 | 60.8 | 74.2 | 77.6 | 83.3 | 80.4 | 48.0 |
| H/14 | I-JEPA | 35.1 | 47.9 | 59.9 | 73.3 | 79.5 | 79.3 | 71.6 |
| | MAE-CT | 49.4 | 59.6 | 67.4 | 74.4 | 81.7 | 82.3 | 79.1 |
| | MAE-Refined | 59.5 | 68.5 | 73.8 | 77.4 | 82.1 | 83.7 | **82.3** |
| | dBOT-Refined | 59.2 | 67.6 | 72.9 | 77.3 | 82.5 | 84.0 | 82.0 |
| | D2V2-Refined | **64.2** | **71.3** | **75.5** | **78.1** | **83.5** | **84.7** | **82.3** |
| 2B/14 | MAE | 17.8 | 29.1 | 62.9 | 73.6 | 82.0 | 79.7 | 67.1 |
| | MAE-Refined | **58.2** | **68.6** | **74.8** | **78.7** | **82.5** | **84.5** | **83.2** |
| L/16 | iBOT (14M) | 37.4 | 49.9 | 61.9 | 70.9 | 80.3 | 82.7 | 72.9 |
| L/7 | MSN (1.3M) | 57.1 | 66.4 | 72.1 | 75.1 | - | 80.7 | - |
| H/14 | D2V2-Refined (1.3M) | **64.2** | **71.3** | **75.5** | 78.1 | 83.5 | 84.7 | 82.3 |
| g/14 | DINOv2 (142M) | 60.5 | 68.3 | 74.4 | **79.1** | **83.8** | **86.5** | **83.5** |

## B.7 EXTENDED IMAGENET-1K CLUSTER EVALUATIONS

Table 11 extends the main paper results (Table 3) with additional models. Table 12 shows the average clustering results of the 100 mini-batch $k$-means runs as described in Section 4.3. We see that MIM-Refiner has also better average performance than competitor methods.

Table 11: Best $k$-means cluster performance and class separation on ImageNet of recent SSL models. This table extends Table 3 from the main paper with additional comparison and refined models.

| ViT | Method | Cluster Performance | | | | Class Separation | |
|---|---|---|---|---|---|---|---|
| | | ACC | NMI | AMI | ARI | SIL ($\uparrow$) | DBS ($\downarrow$) |
| L/16 | MAE | 18.5 | 55.2 | 29.1 | 6.9 | -5.9 | 5.0 |
| | D2V2 | 10.5 | 45.1 | 19.5 | 2.5 | -9.1 | 6.4 |
| | iBOT | 52.2 | 80.5 | 67.0 | 33.4 | 13.3 | 3.5 |
| | Mugs | 54.3 | 78.6 | 65.5 | 22.4 | 14.9 | 3.3 |
| | MAE-Refined | 61.8 | 84.0 | 72.6 | **40.7** | 21.4 | 2.9 |
| | D2V2-Refined | **67.4** | **86.3** | **76.2** | 40.5 | **37.1** | **2.2** |
| H/14 | MAE | 14.3 | 50.2 | 24.2 | 4.3 | -7.8 | 5.2 |
| | D2V2 | 9.9 | 45.8 | 18.0 | 2.6 | -10.8 | 6.5 |
| | MAE-Refined | 64.6 | 85.3 | 74.6 | **45.5** | 21.0 | 2.9 |
| | D2V2-Refined | **67.3** | **87.2** | **77.9** | 42.2 | **34.5** | **2.3** |
| 2B/14 | MAE | 19.9 | 54.1 | 33.1 | 6.2 | -3.6 | 4.8 |
| | MAE-Refined | **63.0** | **85.0** | **74.4** | **44.0** | **14.0** | **3.2** |
| g/14 | DINOv2 | 47.7 | 76.3 | 63.6 | 5.1 | 30.4 | 2.8 |

Table 12: Average $k$-means cluster performance on ImageNet of recent SSL models. MIM-Refiner drastically improves performance of unrefined models and outperforms competitors.

| ViT | Method | Cluster Performance | | | |
|---|---|---|---|---|---|
| | | ACC | NMI | AMI | ARI |
| L/16 | MAE | 17.9 | 54.5 | 28.9 | 6.5 |
| | D2V2 | 10.2 | 44.5 | 19.3 | 2.3 |
| | iBOT | 50.5 | 80.0 | 66.6 | 31.6 |
| | Mugs | 50.7 | 77.4 | 64.4 | 18.1 |
| | MAE-Refined | **60.6** | 83.5 | 71.9 | **35.0** |
| | D2V2-Refined | **60.6** | **83.9** | **72.9** | 30.0 |
| H/14 | MAE | 13.8 | 49.8 | 24.4 | 4.2 |
| | D2V2 | 9.7 | 45.2 | 18.1 | 2.5 |
| | MAE-Refined | **63.2** | **84.7** | 74.0 | **40.1** |
| | D2V2-Refined | 60.4 | 84.5 | **74.3** | 28.4 |
| 2B/14 | MAE | 19.2 | 53.5 | 32.9 | 5.7 |
| | MAE-Refined | **59.4** | **84.0** | **73.4** | **38.0** |
| g/14 | DINOv2 | 46.8 | 75.6 | 62.7 | 3.5 |

Figure 8 shows an additional analysis on the cluster structure in each of the blocks for the refined and unrefined MAE and D2V2 models with ViT-H/14 on ImageNet-1K. Corresponding to the $k$-NN accuracy analysis in Figure 3b, we see that MAE and D2V2 have a higher cluster accuracy (ACC) in the intermediate blocks than in the last block. MIM-Refiner turns this behaviour around and causes a steep increase of ACC starting from the intermediate block and continuing to almost 70% in the last layer. The silhouette score (SIL) confirms this as well. The early blocks allow no separation of the ground truth ImageNet-1K classes leading to negative SIL values, whereas later blocks of the refined models increase separation by 30-50% compared to unrefined counterparts. Interestingly, MAE-Refined is not plateauing in SIL and ACC compared to D2V2-Refined, pointing to potential room for improvement in MAE refinement by using additional ID heads at earlier layers. The bottom part of Figure 8 measures the pairwise cluster label similarity between subsequent blocks in terms of normalized mutual information (NMI) (Kvalseth, 1987) as $(\mathrm{NMI}(y_i, y_{i+1})) \cdot 100$, where $y_i$ are the $k$-means cluster labels at block $i$. The low cluster label similarity in the early blocks for all models indicates that the found cluster labels focus on different clusterings. The later blocks of MIM-Refined models have high alignment with similarities of more than 90%. The higher ACC w.r.t. the ground truth ImageNet-1K classes in the later blocks indicates that they focus more on object-centric features. This is in contrast to the overall lower cluster label similarity in unrefined models, which focus on different cluster structures in each block.

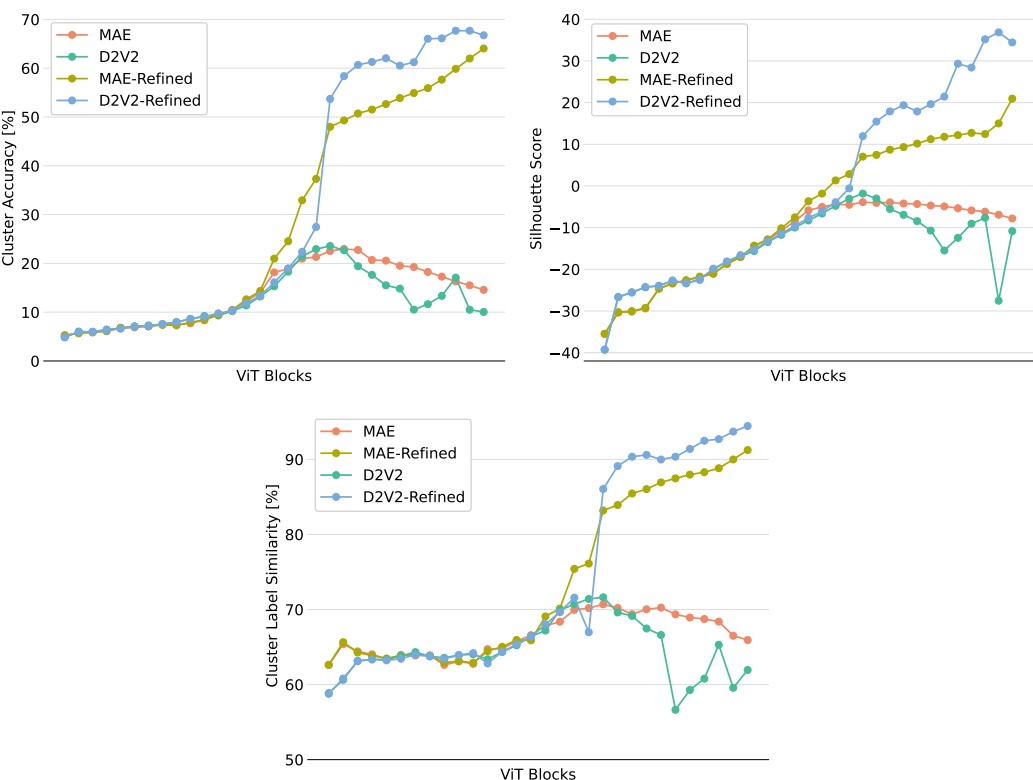

Figure 8: Block-wise cluster analysis for refined and unrefined MAE and D2V2 (H/14). **Upper Left**: Cluster accuracy w.r.t. lowest $k$-means loss per block. **Upper Right**: Silhouette score w.r.t. ground truth ImageNet-1K classes. **Bottom**: Pairwise cluster label similarity between subsequent blocks (details in Section B.7).

## B.8 MULTI-MODEL CLUSTERING EVALUATIONS

We evaluate unsupervised cluster accuracy of MIM-Refiner in combination with other foundation models using the TURTLE (Gadetsky et al., 2024) framework. The results in Table 13 show that MIM-Refiner learns features that are complementary to features learned from foundation models. Combining multiple MIM-Refiner models boosts unsupervised classification accuracy of individual models surpassing the best $k$-means accuracy of D2V2-Ref.-H/14 (67.3%). TURTLE with the feature spaces of MAE-Ref., dBOT-Ref. and D2V2-Ref improves the state-of-the-art of unsupervised classification accuracy using only ImageNet-1K for pre-training to 71.6%. When additionally including foundation models that were pre-trained on web-scale data, MIM-Refiner consistently improves performance. MAE-Refined in combination with DINOv2 (Oquab et al., 2023), CLIP (Radford et al., 2021) and SWAG (Singh et al., 2022) sets a new state-of-the-art of 76.8%.

We conduct this study by implementing MIM-Refiner models into the official implementation of TURTLE[1] and train with the recommended default settings. We do not tune any hyperparameters.

Table 13: Unsupervised classification evaluation using the TURTLE (Gadetsky et al., 2024) framework. MIM-Refiner models synergize well with foundation models such as DINOv2 (Oquab et al., 2023), CLIP (Radford et al., 2021) and SWAG (Singh et al., 2022). MIM-Refiner in combination with TURTLE (Gadetsky et al., 2024) sets a new state-of-the-art in ImageNet-1K unsupervised classification without additional data (71.6%) and with additional data (76.8%). Model sizes are H/14 for MIM-Refiner and SWAG (Singh et al., 2022), g/14 for DINOv2 (Oquab et al., 2023) and L/14 for CLIP (Radford et al., 2021).

| ImageNet-1K pre-training | | | Web-scale pre-training | | | |
|---|---|---|---|---|---|---|
| MAE-Ref. | dBOT-Ref. | D2V2-Ref. | DINOv2 | CLIP | SWAG | ACC |
| *MIM-Refiner only* | | | | | | |
| ✗ | ✓ | ✓ | ✗ | ✗ | ✗ | 61.7 |
| ✓ | ✗ | ✓ | ✗ | ✗ | ✗ | 62.2 |
| ✓ | ✓ | ✗ | ✗ | ✗ | ✗ | 70.4 |
| ✓ | ✓ | ✓ | ✗ | ✗ | ✗ | **71.6** |
| *MIM-Refiner + DINOv2* | | | | | | |
| ✗ | ✗ | ✗ | ✓ | ✗ | ✗ | 68.5 |
| ✗ | ✗ | ✓ | ✓ | ✗ | ✗ | 58.3 |
| ✗ | ✓ | ✗ | ✓ | ✗ | ✗ | **74.0** |
| ✓ | ✗ | ✗ | ✓ | ✗ | ✗ | 73.7 |
| *MIM-Refiner + DINOv2 + CLIP* | | | | | | |
| ✗ | ✗ | ✗ | ✓ | ✓ | ✗ | 72.9 |
| ✗ | ✗ | ✓ | ✓ | ✓ | ✗ | 69.4 |
| ✗ | ✓ | ✗ | ✓ | ✓ | ✗ | **75.0** |
| ✓ | ✗ | ✗ | ✓ | ✓ | ✗ | **75.0** |
| *MIM-Refiner + DINOv2 + CLIP + SWAG* | | | | | | |
| ✗ | ✗ | ✗ | ✓ | ✓ | ✓ | 74.8 |
| ✗ | ✗ | ✓ | ✓ | ✓ | ✓ | 74.9 |
| ✗ | ✓ | ✗ | ✓ | ✓ | ✓ | 76.4 |
| ✓ | ✗ | ✗ | ✓ | ✓ | ✓ | **76.8** |

---

[1]https://github.com/mlbio-epfl/turtle

## B.9 COCO OBJECT DETECTION AND INSTANCE SEGMENTATION

We fine-tune MIM models and their refined versions on COCO (Lin et al., 2014) using the Mask R-CNN (He et al., 2017) configuration of the ViTDet (Li et al., 2022) framework from detectron2[2]. We train for 100 epochs using a batch size of 64, a peak learning rate of 1e-4 with the AdamW (Kingma & Ba, 2015; Loshchilov & Hutter, 2019) optimizer and a multi-step schedule. The results in Table 14 further underline that refined models are competitive with unrefined models on dense downstream tasks. Note that MIM models are extremely good at this type of downstream task. For example, training a pre-trained MAE-H further via weakly-supervised training on a web-scale dataset of 3 billion images even degraded performance vs a plain ImageNet-1K pre-trained MAE by 1.0 $AP^{box}$ and 1.3 $AP^{mask}$ (Singh et al., 2023).

| Model | MAE $AP^{box}$ | MAE $AP^{mask}$ | dBOT $AP^{box}$ | dBOT $AP^{mask}$ | D2V2 $AP^{box}$ | D2V2 $AP^{mask}$ |
|---|---|---|---|---|---|---|
| MIM | **55.2** | **49.1** | **54.7** | **48.7** | **56.0** | **49.5** |
| MIM-Refiner | 54.9 | 48.8 | 54.6 | 48.6 | 55.5 | 49.3 |

Table 14: Fine-tuning results for L/16 models on COCO object detection and instance segmentation.

## B.10 EXTENDED TRANSFER LEARNING RESULTS

Table 15 extends Table 4 with additional results and comparison to more SSL models.

Table 15: Extending Table 4 with additional MIM-Refiner models and additional SSL models. MIM-Refiner learns general features that can easily be transferred to various datasets and tasks.

| ViT | Method | iNat18 Fine-tuning 1-shot | 5-shot | 10-shot | Linear Probe iNat18 | VTAB-6 | ADE20K |
|---|---|---|---|---|---|---|---|
| L/16 | CrossMAE | 5.4 | 45.6 | 63.7 | 39.8 | 81.2 | 30.9 |
| | MAE | 7.1 | 51.6 | 68.9 | 42.8 | 83.0 | 33.6 |
| | dBOT | 7.1 | 53.1 | 71.2 | 44.6 | 83.4 | 34.1 |
| | D2V2 | 5.5 | 53.1 | 70.6 | 39.4 | 81.5 | 38.8 |
| | CAE | 7.4 | 54.5 | 71.2 | 46.4 | 84.8 | 36.5 |
| | iBOT | 15.8 | 51.5 | 65.5 | 56.0 | 87.6 | 35.6 |
| | Mugs | **19.5** | 53.2 | 66.9 | 61.5 | 87.9 | 34.8 |
| | CrossMAE-Ref | 18.0 | 52.9 | 67.8 | 60.5 | 88.3 | 34.3 |
| | MAE-Ref | 19.0 | 58.0 | 71.7 | 60.6 | **88.5** | 37.3 |
| | dBOT-Ref | 18.3 | **58.8** | **73.0** | **61.7** | **88.5** | 38.4 |
| | D2V2-Refined | 15.2 | 56.3 | 71.8 | 52.0 | 85.9 | **41.0** |
| H/14 | MAE | 6.5 | 53.3 | 71.7 | 43.0 | 83.2 | 35.5 |
| | dBOT | 6.8 | 53.1 | 73.2 | 46.2 | 84.6 | 36.1 |
| | D2V2 | 5.9 | 55.7 | 73.4 | 41.7 | 83.1 | 42.4 |
| | MAE-CT | 16.5 | 60.1 | 74.7 | 62.8 | 88.8 | 37.6 |
| | MAE-Ref | **20.9** | **62.4** | 75.4 | 64.6 | 89.3 | 39.4 |
| | dBOT-Ref | 20.0 | 60.9 | **75.8** | **65.8** | **89.5** | 37.6 |
| | D2V2-Refined | 16.1 | 59.2 | 74.8 | 54.4 | 87.1 | **43.7** |
| 2B/14 | MAE | 10.0 | 53.7 | 72.2 | 51.0 | 85.4 | 37.3 |
| | MAE-Ref | **22.5** | **63.5** | **76.5** | **69.6** | **89.8** | **40.3** |
| L/7 | MSN | 17.0 | 38.0 | 48.1 | - | - | - |

[2]https://github.com/facebookresearch/detectron2

### B.11 VTAB INDIVIDUAL DATASET RESULTS

We show the individual accuracies for each VTAB (Zhai et al., 2019) dataset from Table 4 of the main paper. Table 16 shows linear probing results on VTAB and Table 17 shows fine-tuning results on VTAB-1K. We use only six datasets for linear probing, as the other datasets are quite different to the ImageNet-1K images seen during training and therefore benefit heavily from fine-tuning which makes them more suited for evaluation via fine-tuning instead of linear probing.

Table 16: Linear probing accuracy on six VTAB (Zhai et al., 2019) datasets from the "Natural" category: Caltech101 (Fei-Fei et al., 2006), CIFAR100 (Krizhevsky, 2009), DTD (Cimpoi et al., 2014), Flowers102 (Nilsback & Zisserman, 2008), Pets (Parkhi et al., 2012) and Sun397 (Xiao et al., 2010).

| ViT | Method | VTAB Dataset | | | | | | |
|---|---|---|---|---|---|---|---|---|
| | | CF100 | CT101 | DTD | FL102 | Pets | Sun397 | Average |
| L/16 | CrossMAE | 81.4 | 88.8 | 75.6 | 84.2 | 84.1 | 72.7 | 81.2 |
| | MAE | 80.0 | 91.8 | 75.6 | 86.3 | 89.6 | 74.7 | 83.0 |
| | dBOT | 84.3 | 91.3 | 75.7 | 88.0 | 86.0 | 75.1 | 83.4 |
| | D2V2 | 85.0 | 89.8 | 74.0 | 84.9 | 80.7 | 74.6 | 81.5 |
| | CAE | 87.0 | 92.3 | 76.1 | 88.4 | 89.2 | 75.9 | 84.8 |
| | iBOT | 89.3 | 91.3 | 78.1 | 95.8 | 93.7 | 77.1 | 87.6 |
| | Mugs | 89.5 | 90.5 | 78.0 | **96.8** | 95.3 | 77.2 | 87.9 |
| | CrossMAE-Ref | 88.7 | **93.5** | **79.0** | 95.7 | 95.2 | 78.1 | 88.3 |
| | MAE-Ref | 89.1 | 91.9 | **79.0** | 96.2 | **95.8** | 78.8 | **88.5** |
| | dBOT-Ref | **90.4** | 91.3 | 78.6 | 95.9 | 95.5 | **79.2** | **88.5** |
| | D2V2-Ref | 88.9 | 88.8 | 73.9 | 92.1 | 94.7 | 77.1 | 85.9 |
| H/14 | MAE | 81.0 | 90.3 | 76.9 | 85.9 | 89.5 | 75.3 | 83.2 |
| | dBOT | 85.5 | 91.7 | 77.7 | 88.1 | 88.2 | 76.3 | 84.6 |
| | D2V2 | 87.1 | 91.4 | 77.4 | 85.9 | 79.9 | 76.9 | 83.1 |
| | I-JEPA | 87.1 | 92.7 | 72.5 | 90.4 | 92.4 | 74.9 | 85.0 |
| | MAE-CT | 87.7 | **93.9** | 80.1 | 97.0 | 95.0 | 79.2 | 88.8 |
| | MAE-Ref | 90.1 | 92.0 | 80.4 | **97.5** | 96.0 | 79.8 | 89.3 |
| | dBOT-Ref | **91.7** | 92.1 | **80.6** | 96.7 | **96.1** | **80.1** | **89.5** |
| | D2V2-Ref | 90.4 | 89.0 | 75.9 | 93.3 | 95.6 | 78.4 | 87.1 |
| 2B/14 | MAE | 82.5 | 92.0 | 78.2 | 90.5 | 91.8 | 77.1 | 85.4 |
| | MAE-Ref | **90.8** | **92.6** | **81.1** | **97.7** | **96.5** | **80.3** | **89.8** |

Table 17: Fine-tuning accuracy of all 19 VTAB-1K (Zhai et al., 2019) datasets (averages are reported in Table 4). Row-groups correspond to ViT-L/16, ViT-H/14 and ViT-2B/14 models respectively.

| Method | Natural | | | | | | | Specialized | | | | Structured | | | | | | | |
|---|---|---|---|---|---|---|---|---|---|---|---|---|---|---|---|---|---|---|---|
| | CF100 | CT101 | DTD | FL102 | Pets | SVHN | Sun397 | Camelyon | EuroSAT | Resisc45 | Retinopathy | Clevr-Count | Clevr-Dist | DMLab | KITTI-Dist | dSpri-Loc | dSpr-Ori | sNORB-Azi | sNORB-Ele |
| MAE | 50.3 | 90.1 | 68.0 | 90.8 | 92.7 | **92.2** | 35.1 | **85.5** | 93.2 | 81.7 | 72.7 | **93.2** | 63.9 | 55.8 | **84.5** | **93.9** | 58.6 | **46.1** | 51.9 |
| iBOT | 68.3 | **91.5** | **71.5** | 95.8 | 91.8 | 80.1 | 47.6 | 85.4 | 93.4 | 86.6 | 74.5 | 79.9 | 61.7 | 52.3 | 80.2 | 74.5 | 47.2 | 27.5 | 35.4 |
| Mugs | **69.4** | 89.2 | 71.4 | 95.4 | 92.9 | 78.1 | 47.6 | 85.2 | 94.0 | 86.2 | **74.7** | 75.1 | 53.0 | 51.1 | 77.9 | 46.8 | 45.5 | 26.3 | 32.6 |
| MAE-Ref | 62.2 | 90.6 | **71.5** | **97.2** | **93.7** | 91.7 | **49.1** | 85.3 | **94.8** | **86.8** | 73.4 | 92.5 | **64.3** | **57.4** | 84.3 | 88.4 | **60.5** | 39.0 | 46.5 |
| MAE | 44.3 | 88.3 | 67.8 | 90.8 | 90.8 | **91.1** | 30.4 | 85.9 | 94.0 | 81.3 | 73.9 | 92.7 | 63.6 | 54.0 | 84.9 | 92.2 | 62.8 | 37.1 | **55.2** |
| MAE-CT | 58.5 | **93.2** | 72.3 | 97.5 | 93.4 | 90.7 | 48.3 | 85.9 | 94.4 | 86.1 | 75.0 | **93.0** | 64.9 | 58.3 | 83.6 | 91.9 | **63.8** | **38.0** | 48.7 |
| MAE-Ref | 62.3 | 91.3 | 73.1 | 97.9 | 93.9 | 89.5 | 49.7 | **86.8** | **94.9** | **87.7** | **75.1** | 92.9 | 63.3 | 57.5 | **85.9** | **93.5** | 62.2 | 36.0 | 49.3 |
| MAE | 44.7 | 90.4 | 70.0 | 91.8 | 92.4 | **92.3** | 33.8 | 85.9 | 94.2 | 82.6 | **74.2** | 92.4 | 63.9 | 55.5 | **85.5** | **94.6** | 61.1 | **44.9** | **57.9** |
| MAE-Ref | **61.8** | **91.0** | **73.9** | **97.5** | **94.5** | 91.3 | **49.1** | **86.6** | **94.7** | **87.0** | 73.8 | **92.6** | **64.7** | **58.8** | 85.3 | 93.5 | **61.2** | 37.9 | 40.9 |

B.12    EXTENDED COMPARISON OF FINE-TUNING ON IMAGENET-1K AND INAT18

Table 18 extends Table 5 with additional MIM models. MIM-Refiner consistently improves also on fine-tuning with an abundance of labels. Table 19 shows individual performances on robustness and domain generalization benchmarks.

Table 18: Full fine-tuning using 100% of the labels. MIM-Refiner consistently improves performance slightly even though ID methods typically perform worse than MIM models on this benchmark. This table extends Table 5 from the main paper with additional MIM models and SSL models.

| Model | ViT-L/16 | | ViT-H/14 | |
|---|---|---|---|---|
| | ImageNet-1K | iNat18 | ImageNet-1K | iNat18 |
| MAE | **85.7** | 80.7 | 86.7 | 82.7 |
| MAE-CT | 85.4 | **80.9** | 86.8 | 82.9 |
| MAE-Refined | 85.6 | **80.9** | **86.9** | **83.3** |
| CrossMAE | 84.9 | 77.7 | - | - |
| CrossMAE-Refined | **85.1** | **78.4** | - | - |
| D2V2 | 86.6 | 81.0 | 86.6 | 79.6 |
| D2V2-Refined | **86.7** | **81.6** | **86.8** | **79.8** |
| dBOT | 85.8 | 81.9 | 87.1 | 84.1 |
| dBOT-Refined | **85.9** | **82.1** | 87.1 | **84.5** |
| iBOT | 84.8 | 76.9 | - | - |
| Mugs | 85.2 | 76.9 | - | - |
| I-JEPA | - | - | 84.9 | 75.9 |

Table 19: Robustness and domain generalization evaluation on ImageNet-C(orruption) (Hendrycks & Dietterich, 2019) ImageNet-A(dversarial) (Hendrycks et al., 2021b), ImageNet-R(endition) (Hendrycks et al., 2021a) and ImageNet-Sketch (Wang et al., 2019b). For ImageNet-C we report the mean corruption error (Hendrycks & Dietterich, 2019). MIM-Refiner consistently improves robustness, particularly on larger models. Table 5 reports the average accuracy of IN-A, IN-R and IN-Sketch.

| ViT | Method | IN-C (↓) | IN-A (↑) | IN-R (↑) | Sketch (↑) | Validation (↑) |
|---|---|---|---|---|---|---|
| L/16 | CrossMAE | 41.0 | 51.8 | **57.1** | 42.0 | 84.9 |
| | CrossMAE-Ref. | **40.7** | **52.1** | 57.0 | **42.1** | **85.1** |
| L/16 | MAE | 39.2 | 56.2 | 60.0 | 45.6 | **85.7** |
| | MAE-Ref. | **39.0** | **57.0** | **60.1** | **45.9** | 85.6 |
| L/16 | D2V2 | 34.0 | 66.1 | **64.4** | 50.0 | 86.6 |
| | D2V2-Ref. | **33.5** | **67.2** | 64.3 | **50.1** | **86.7** |
| L/16 | dBOT | **36.1** | 60.0 | **60.5** | **45.5** | 85.8 |
| | dBOT-Ref. | **36.1** | **60.3** | 60.4 | 45.3 | **85.9** |
| L/16 | iBOT | 37.8 | 47.6 | 53.7 | 41.9 | 84.5 |
| | Mugs | 39.9 | 47.8 | 52.0 | 39.3 | 84.6 |
| H/14 | MAE | 34.6 | 67.8 | 64.1 | 48.9 | 86.7 |
| | MAE-Ref. | **34.3** | **68.0** | **65.1** | **49.7** | **86.9** |
| H/14 | D2V2 | 30.7 | 72.9 | 65.7 | 51.0 | 86.6 |
| | D2V2-Ref. | **30.2** | **74.1** | **66.1** | **52.1** | **86.8** |
| H/14 | dBOT | **31.8** | 71.1 | **68.1** | **51.8** | **87.1** |
| | dBOT-Ref. | **31.8** | **72.3** | 68.0 | **51.8** | **87.1** |
| H/14 | I-JEPA | 37.4 | 51.7 | 57.4 | 43.7 | 81.7 |
| 2B/14 | MAE | 32.6 | 68.4 | 65.4 | 50.0 | 86.7 |
| | MAE-Ref. | **32.2** | **68.9** | **66.2** | **50.5** | **86.8** |

## B.13 FINE-TUNING VIT-2B MODELS

Table 20 shows results for fine-tuning ViT-2B models.

Table 20: Fine-tuning ViT-2B models using 100% of the labels. As fine-tuning these models is expensive, we freeze the first 6 of the 24 blocks to save memory and compute.

| | ViT-2B/14 | |
|---|---|---|
| Model | ImageNet-1K | iNat18 |
| MAE | 86.7 | 82.2 |
| MAE-Refined | **86.9** | **83.2** |

## B.14 IMPACT OF MULTI-CROP AUGMENTATION

Multi-crop augmentation (Caron et al., 2020) has been shown to greatly improve the performance of ID methods (Caron et al., 2020; 2021; Zhou et al., 2021; 2022) and also improves MIM-Refiner significantly, where the $k$-NN accuracy of a D2V2-Refined L/16 drops by 2.6% when omitting multi-crop augmentations. When comparing to drops of other models, this is a relatively small drop. For example, the performance of DINO B/16 drops by 7.2% and iBOT B/16 drops by 5.6% when omitting multi-crop augmentations (see Table 10 in (Zhou et al., 2021)).

## B.15 HIGH-DIMENSIONAL $k$-NN

The linear probing protocol of DINOv2 (Oquab et al., 2023) includes the possibility to concatenate features of the last 4 blocks as input to the linear probe. We find that this outperforms using only the features of the last block in most cases. As the best linear probe uses features from the last 4 blocks and the fact that the $k$-NN and linear probing metrics are typically correlated (Oquab et al., 2023), we report the $k$-NN accuracy using only the features of the last block in Table 2 to use the last 4 blocks for one metric and only the last block for the other. In Table 21 we investigate using the concatenation of features from the last 4 blocks for a $k$-NN classifier. Most models benefit from using more features, especially MIM models.

Table 21: ImageNet-1K $k$-NN accuracy at 224x224 resolution of the [CLS] token of the last block or the concatenation of the [CLS] tokens of the last 4 blocks.

| | | #Blocks | | |
|---|---|---|---|---|
| ViT | Method | 1 | 4 | Delta |
| | MAE | 60.6 | 63.3 | +2.7 |
| | D2V2 | 51.8 | 52.9 | +1.1 |
| | iBOT | 78.0 | 78.9 | +0.9 |
| L/16 | Mugs | 80.4 | 80.1 | -0.3 |
| | MAE-Refined | **81.5** | 81.5 | 0.0 |
| | D2V2-Refined | 81.0 | **81.7** | +0.7 |
| | MAE | 58.1 | 61.4 | +3.3 |
| | D2V2 | 48.0 | 52.2 | +4.2 |
| H/14 | I-JEPA | 71.6 | 72.3 | +0.7 |
| | MAE-CT | 79.1 | 78.6 | -0.5 |
| | MAE-Refined | **82.3** | 82.5 | +0.2 |
| | D2V2-Refined | **82.3** | **83.4** | +1.1 |
| g/14 | DINOv2 | 83.0 | 83.9 | +0.9 |

## B.16 MIM-REFINER ON SMALLER MODELS

MIM-Refiner builds on pre-trained MIM models which excel at larger scales (ViT-L and upwards), we mainly focus on large-scale models in our paper. However, MIM-Refiner also improves MIM

models on smaller scales (ViT-B). Additionally, combinations of MIM and ID methods have been explored in various works (Huang et al., 2022b; Wang et al., 2022; Yi et al., 2023). However, as these methods introduce significant runtime overhead over MIM models, they mainly focus on smaller models (ViT-L and smaller) where the pre-trained models are also often not published, which makes a comprehensive comparison against these models impossible. Nevertheless, we show that MIM-Refiner is complementary to these methods by refining a Contrastive MAE (Huang et al., 2022b) ViT-B/16 model with MIM-Refiner. Table 22 shows that MIM-Refiner also significantly improves representation quality on smaller models.

Table 22: MIM-Refiner also significantly improves ViT-B models. Methods that improve MIM by incorporating ID already during pre-training are orthogonal to MIM-Refiner where the refinement process also significantly improves representation quality of these models.

| ViT-B/16 | $k$-NN | 5-shot | 2-shot | 1-shot |
|---|---|---|---|---|
| MAE | 51.1 | 43.1 | 27.1 | 14.0 |
| MAE-Refined | **76.6** | **64.5** | **58.6** | **50.0** |
| CMAE | 76.7 | 43.3 | 31.2 | 21.7 |
| CMAE-Refined | **78.5** | **70.1** | **65.7** | **57.9** |

### B.17 RUNTIME OVERHEAD OF ID QUEUE

We benchmark the overhead of the queue with its topk lookup in Table 23. The queue only adds a small amount of overhead as it is only needed for the forward pass (not for the backward pass) and the queue operates in the bottleneck dimension (256 for all models) of the ID head. Our setup for the queue closely follows the one introduced in Dwibedi et al. (2021). We estimate the runtime of three configurations where the last one is used for training MIM-Refiner models: (i) no queue (ii) queue with top1 NN-swap and (iii) queue with top20 NN-swap. We train these configurations for a short amount of time on a single GPU with the maximal possible batchsize and report the average runtime per sample. Note that the overhead is so small that random runtime fluctuations that are common in modern GPU setups can slightly distort the results.

| Queue size | topk | L/16 | H/14 | 2B/14 |
|---|---|---|---|---|
| 0 | - | 12.8s | 30.1s | 76.7s |
| 65K | 1 | 12.9s | 30.2s | 76.9s |
| 65K | 20 | 13.0s | 30.5s | 77.5s |

Table 23: Runtime per sample for different queue configurations. The queue adds only a small amount of overhead. MIM-Refiner uses a queue size of 65K with a top20 NN lookup.

## C IMPLEMENTATION DETAILS

### C.1 EVALUATIONS

To avoid slight performance differences due to minor implementation details or version changes and facilitate a fair comparison between models, we run all evaluations on our own in accordance with the suggested hyperparameters of the respecitve methods.

### C.2 HARDWARE

All models are pre-trained on multiple nodes of 4xA100-64GB GPUs where ViT-L uses 4 nodes (i.e. 16 GPUs), ViT-H 8 nodes of 4xA100 (i.e. 32 GPUs) and ViT-2B uses 16 nodes (i.e. 64 GPUs). For evaluations, we use a mix of 4xA100-64GB nodes, 8xA100-40GB nodes and various smaller nodes that vary in number of GPUs. We estimate the total number of A100 GPU-hours used for this project to be 40K hours. This estimate includes everything from initial exploration, method development, analysis and evaluations.

## C.3 VISION TRANSFORMER

The architecture of our models follows the ones from the respective MIM model that is refined. That is a pre-norm architecture for MAE (He et al., 2022) and a post-norm architecture for data2vec 2.0 (Baevski et al., 2023). We attach the ID heads to the [CLS] tokens and also use the [CLS] token for evaluation.

We download the official checkpoints from the respective MIM works. Note that the official Cross-MAE model is pre-trained for less epochs than all other MIM models. For dBOT we use the models where a teacher of the same size is used (i.e. dBOT-L used MAE-L as teacher and dBOT-H used MAE-H as teacher).

## C.4 ID HEAD ARCHITECTURE

We use a three layer MLP with hidden dimension 2048 as projector and a two layer MLP with hidden dimension 4096 as predictor. Each linear projection is followed by a GELU (Hendrycks & Gimpel, 2016) and a batchnorm (Ioffe & Szegedy, 2015) layer. For the last linear projection in projector and predictor, no GELU is used.

## C.5 REFINEMENT HYPERPARAMETERS

Hyperparameters for the refinement stage are listed in Table 24. Following MAE-CT (Lehner et al., 2024), we initialize all ID heads first by training them with a frozen encoder to ensure a good learning signal from the start of the refinement process. For this initialization, we use the same hyperparameters as in Table 24 except that we use 20 epochs for all models, a learning rate of 2e-4 and a top1-NN lookup. As we do not use a momentum encoder during training, we instead track an EMA of the encoder and use the EMA then for downstream tasks. As ViT-2B is very expensive to train, we freeze the first 6 blocks (for refinement and also for evaluation). As shown in Table 8 this slightly reduces performance but also reduces memory consumption and runtime.

Table 24: MIM-Refiner hyperparameters.

| Parameter | Value | Parameter | Value |
|---|---|---|---|
| Epochs | 30 (MAE/dBOT L/H) 20 (MAE 2B, data2vec 2.0) | NNCLR Heads | |
| | | Weight Decay | 1e-5 |
| Batch Size | 1024 (L), 512 (H, 2B) | Temperature | 0.2 (L), 0.3 (H), 0.35 (2B) |
| Optimizer | AdamW | topk-NN $k$ | 20 |
| Learning Rate | 4e-4 | NN-swap for Positives | ✓ |
| Momentum | $\beta_1 = 0.9, \beta_2 = 0.95$ | NN-swap for Negatives | ✗ |
| Learning Rate Schedule | Linear Warmup → Cosine Decay | Data Augmentation | |
| Warmup Epochs | 4 | Color & Blur Settings | see BYOL |
| End Learning Rate | 1e-6 | Global Views | 2 |
| Encoder | | Global View Resolution | 224 |
| Layer-wise LR Decay | 0.65 | Global View Scale | [0.25, 1.0] |
| Freeze Blocks | 0 (L/H), 6 (2B) | Local Views | 10 |
| Weight Decay | 0.05 | Local View Resolution | 96 (L), 98 (H, 2B) |
| EMA | 0.9999 | Local View Scale | [0.05, 0.25] |

# D EVALUATION DETAILS

## D.1 GPU HOURS BENCHMARK

For benchmarking GPU hours, we follow the setup from MAE-CT (Lehner et al., 2024): we conduct a comparison by implementing all methods in a single code-base and conducting short training runs on a single A100 40GB PCIe card. These runs are executed in mixed precision training mode and with the highest possible batchsize that is a power of 2. The runtime of these benchmark runs is then extrapolated to the reported number of epochs. Benchmarks are conducted in pytorch 2.1 with CUDA 12.1. FLOPS are measured with the fvcore library[3]. For the 1-shot classification plot in Figure 2, we do not take into account that some models train on higher resolutions (e.g. DINOv2) for visual clarity.

---

[3]https://github.com/facebookresearch/fvcore

## D.2 MAE INTERMEDIATE REPRESENTATION ANALYSIS

We analyze how well the features of a ViT block are suited for reconstruction by training an MAE with a decoder attached after every ViT block. We use the same parameters as for training from scratch (He et al., 2022) but reduce training duration to 20 epochs, warmup to 5 epochs and the depth of all decoders to 2. The encoder remains fully frozen during training and only the decoders are trained.

For the visualization in Figure 3d, we calculate the delta from one block to the next. We do this for both the $k$-NN accuracy and the reconstruction loss. Additionally, we divide by the maximum delta of each metric to transform both metrics into a similar value range and upper bound 1. Figure 9 shows the reconstruction loss per block and the same plot for a MAE L/16, where a similar behavior can be observed.

We conduct the same analysis for refined models and show results in Figure 10.

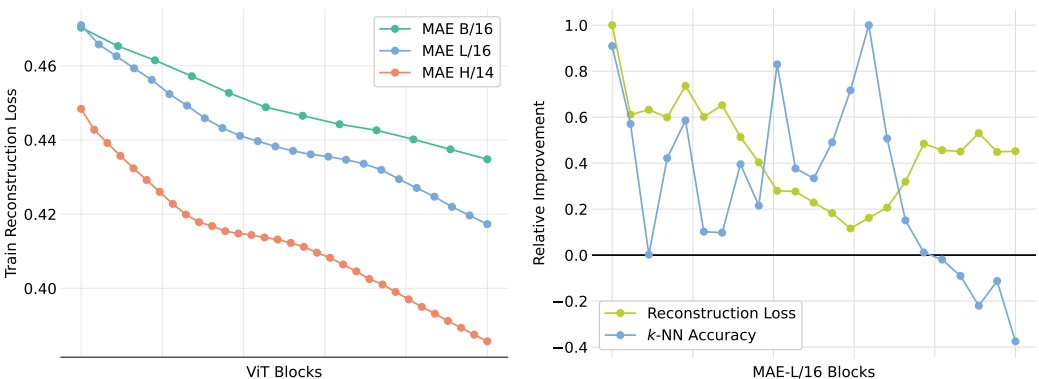

Figure 9: **Left**: Reconstruction loss per block of MAEs. **Right** Relative improvement of reconstruction loss and $k$-NN accuracy for a MAE L/16.

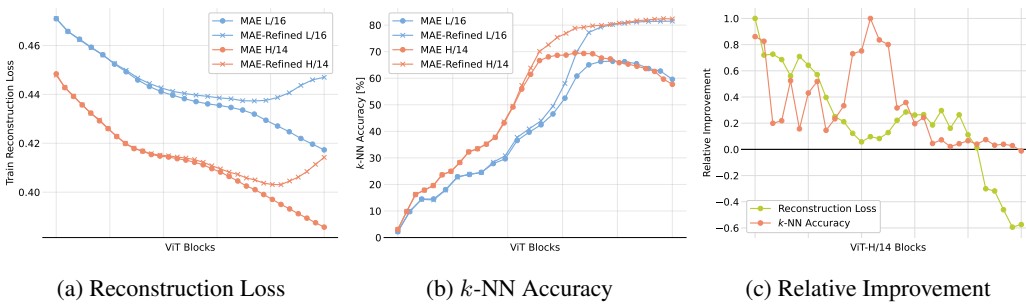

(a) Reconstruction Loss      (b) $k$-NN Accuracy      (c) Relative Improvement

Figure 10: Feature analysis of refined models. (a) The reconstruction loss increases in later blocks after refinement. (b) Representation quality increases drastically in later encoder blocks. (c) The relative improvements of MAE-Refined-H/14 shows that the refinement improves representation quality at the cost of reconstruction loss, which benefits downstream performance.

## D.3 IMAGENET-1K LOW-SHOT EVALUATION DETAILS

For the 1, 2 and 5-shot benchmarks we train a logistic regression (Caron et al., 2021; Assran et al., 2022) using the [CLS] token after the last encoder block with the `cyanure` (Mairal, 2019) library. As MIM models benefit from fine-tuning in this setting (Assran et al., 2022), MAE and data2vec 2.0 are fine-tuned instead. We report the average of three dataset splits from MSN (Assran et al., 2022).

In the 1% and 10% low-shot benchmark, all models are fine-tuned with hyperparameters similar to those used in related works (Lehner et al., 2024; Assran et al., 2023; Cai et al., 2022). As the

parameters vary between 1%/10% and also between model sizes, we refer to the codebase for the exact protocols.

For a fair comparison, we conduct the low-shot evaluations of DINOv2 at 224 resolution. We study the impact of the higher resolution where we observe minimal gains at the original resolution (518x518). Note that DINOv2 first trains at 224x224 followed by a short training at 518x518 resolution.

Table 25: ImageNet-1K low-shot evaluation of DINOv2 g/14 on higher resolutions.

| resolution | #patches | FLOPS [G] | 1-shot | 2-shot | 5-shot |
|---|---|---|---|---|---|
| 224x224 | 256 | 291 | 60.5 | 68.3 | 74.4 |
| 518x518 | 1369 | 1553 | 61.1 | 68.8 | 74.8 |

### D.4 IMAGENET-1K $k$-NN CLASSIFICATION DETAILS

For $k$-NN classification, we follow the protocol of DINO (Wu et al., 2018; Caron et al., 2021). We train a soft $k$-NN classifier weighted by cosine similarity with $k = 10$. For MIM models, higher values for $k$ are beneficial, so we tune this parameter for MAE and data2vec 2.0.

### D.5 IMAGENET-1K LINEAR PROBING EVALUATION DETAILS

For linear probing, we use the protocol from DINOv2 (Oquab et al., 2023) for publicly released models and the values from the original papers otherwise. We train for 50 epochs using SGD with momentum 0.9. As data augmentation we use `RandomResizedCrop` and `HorizontalFlip`. The DINOv2 protocol sweeps over the following hyperparameters by training multiple linear classifiers at once:

- 13 learning rates ranging from 0.0001 to 0.5
- Use the last block output or concatenate the output of the last 4 blocks
- Use the [CLS] token or the concatenation of [CLS] and [AVG] token

As the linear probes trained on the concatenation of the last 4 blocks have more features and more parameters to discriminate between classes, they tend to be the best within the sweeped parameters. Note that we evaluate the representation of the last block in isolation via $k$-NN classification. We investigate $k$-NN classification with features from the last 4 blocks in Appendix B.15.

### D.6 IMAGENET-1K CLUSTER EVALUATION DETAILS

For each considered model in the clustering experiments in Section 4.3 we used the CLS token embeddings of the ImageNet validation set and preprocessed the embeddings using L2 normalization. For conducting mini-batch $k$-means and calculating the cluster related metrics we used the `scikit-learn` package (Pedregosa et al., 2011), except for calculating the cluster accuracy where we used the implementation in `ClustPy` (Leiber et al., 2023). The UMAP plots in Figure 5 where generated by applying UMAP on top of the L2 normalized CLS token embeddings of the 53 food related classes of ImageNet for each model. We use the default UMAP parameters of `umap-learn` (McInnes et al., 2018) for all plots (n_neighbors=15).

### D.7 INAT18 TRANSFER LEARNING EVALUATION DETAILS

We report the accuracy on the validation set averaged over three seeds.

For 1-shot classification on iNat18, we use the linear probing protocol from DINOv2 (Oquab et al., 2023). We also attempted to fine-tune MIM models where some models fail to exceed random performance and therefore also use linear probing.

For 5-shot and 10-shot classification on iNat18, we fine-tune all models. The hyperparameters for fine-tuning (Table 26) are inspired by MAWS (Singh et al., 2023).

Table 26: Hyperparameters for fine-tuning on iNat18 low-shot classification.

| Parameter | Value | Parameter | Value |
|---|---|---|---|
| Epochs | 50 | Label smoothing | 0.1 |
| Batch size | 256 | Data Augmentation | |
| Optimizer | AdamW | `Resize` | 256 |
| Learning rate | 1e-3 | `  interpolation` | bicubic |
| Layer-wise lr decay | 0.75 | `RandomResizedCrop` | 224 |
| Weight decay | 0.05 | `  scale` | [0.08, 1.0] |
| Momentum | $\beta_1 = 0.9, \beta_2 = 0.999$ | `  interpolation` | bicubic |
| Learning rate schedule | linear warmup $\rightarrow$ cosine decay | `RandomHorizontalFlip` | $p = 0.5$ |
| Warmup epochs | 5 | `Normalize` | ImageNet statistics |

### D.8 TRANSFER LEARNING LINEAR PROBING

For transfering the pre-trained features to iNat18, six VTAB datasets and ADE20K (Table 4) we use the DINOv2 (Oquab et al., 2023) linear probing protocol as described in Appendix D.5. For ADE20K and iNat18, we reduce the hyperparameter grid to fit into 40GB GPU memory.

### D.9 ADE20K SEMANTIC SEGMENTATION LINEAR PROBE

Large models (such as ViT-H or ViT-2B) are expensive to train on ADE20K. Therefore, we opt for a simple light-weight evaluation protocol to compare our models on a segmentation task:

- We keep resolution at 224x224
- We freeze the encoder
- We train a linear classifier similar to DINOv2 (Oquab et al., 2023) that predicts a class for each patch. The resulting low-resolution prediction is then upsampled to 224x224 resolution.
- For evaluation, we use the original resolution image and slide a 224x224 window over the image with a stride of 170 pixels and average the logits per pixel.

As intermediate representations are commonly used for semantic segmentation, we use features from the last block, the 5th last block, the 9th last block and the 13th last block. Compared to simply using the last 4 blocks, this improves performance for all compared models.

### D.10 FINE-TUNING ON VTAB-1K

For fine-tuning models on VTAB-1K we provide the hyperparameters in Table 27. We search for the best learning rate for each dataset by fine-tuning the model 25 times (5 learning rates with 5 seeds each) on the 800 training samples and evaluating them on the 200 validation samples. With the best learning rate, we then train each model 5 times on concatenation of training and validation split, evaluate on the test split and report the average accuracy.

Table 27: Hyperparameters for fine-tuning on VTAB-1K.

| Parameter | Value | Parameter | Value |
|---|---|---|---|
| Epochs | 50 | Learning rate schedule | linear warmup $\rightarrow$ cosine decay |
| Batch size | 64 | Warmup epochs | 5 |
| Seeds | 5 | Data Augmentation | |
| Optimizer | AdamW | `Resize` | |
| Learning rate | 1e-3, 7.5e-4, 5.0e-4, 2.5e-4, 1.0e-4 | `  interpolation` | bicubic |
| Layer-wise lr decay | 0.75 | `  size` | 224x224 |
| Weight decay | 0.05 | `Normalize` | ImageNet-1K statistics |
| Momentum | $\beta_1 = 0.9, \beta_2 = 0.999$ | | |

## D.11 FINE-TUNING WITH 100% LABELS

For fine-tuning with 100% of the labels (Table 5), we use the hyperparameters provided in MAE (He et al., 2022) for both iNat18 and ImageNet-1K (see Table 28). As D2V2 models are unstable with the default learning rate of the MAE fine-tuning protocol, we use the highest stable learning rate out of 5e-4, 2.5e-4 and 1e-4.

For ViT-2B/14 (Table 20), we freeze the first 6 of the 24 blocks to reduce computational costs. Additionally, as the 2B models are sometimes unstable with a learning rate of 1e-3, we reduce it to 7.5e-4 or 5e-4 using the largest stable learning rate.

To fine-tune I-JEPA (Assran et al., 2023), we adjust hyperparameters to match their fine-tuning setting of a ViT-H/$16_{448}$. We reduce peak stochastic depth from $0.3$ to $0.25$. To fine-tune on iNat18, we found that a learning rate 1e-3 performs better than the 1e-4 used in the original work.

Table 28: Hyperparameters for fine-tuning on ImageNet-1K and iNat18 many-shot classification.

| Parameter | Value |
| --- | --- |
| Epochs | 50 |
| Batch size | 1024 |
| Stochastic depth | |
|   Peak rate | 0.2 (L/2B), 0.3 (H) |
|   Decay | ✓ |
| Optimizer | AdamW |
|   Learning rate | 1e-3 |
|   Layer-wise lr decay | 0.75 |
|   Weight decay | 0.05 |
|   Momentum | $\beta_1 = 0.9, \beta_2 = 0.999$ |
|   Freeze Blocks | 0 (L/H), 6 (2B) |
| Learning rate schedule | linear warmup $\rightarrow$ cosine decay |
|   Warmup epochs | 5 |
|   End Learning Rate | 1e-6 |
| Label smoothing | 0.1 |

| Parameter | Value |
| --- | --- |
| **Train Data Augmentation** | |
|   RandomResizedCrop | 224 |
|     scale | [0.08, 1.0] |
|     interpolation | bicubic |
|   RandomHorizontalFlip | $p = 0.5$ |
|   RandAug | |
|     magnitude | 9 |
|     magnitude_std | 0.5 |
|   Normalize | ImageNet statistics |
|   Mixup $\alpha$ | 0.8 |
|   Cutmix $\alpha$ | 1.0 |
| **Test Data Augmentation** | |
|   Resize | 256 |
|     interpolation | bicubic |
|   CenterCrop | 224 |
|   Normalize | ImageNet statistics |

## D.12 ADE20K SEMANTIC SEGMENTATION FINE-TUNING

We fine-tune ViT-L models using an UperNet (Xiao et al., 2018) segmentation head to predict a segmentation mask. We follow common practices and fine-tune on 512x512 resolution, where we interpolate the absolute positional embedding from 224x224 to 512x512, add relative position bias to the attention layers (initialized to 0) (He et al., 2022) and introduce layerscale (Touvron et al., 2021) (initialized to 1). A common augmentation pipeline is used that consists of random rescaling, random horizontal flipping, color jitter and padding if necessary. We train for 160K updates using a batchsize of 16, a learning rate of 2e-5, weight decay 0.05, linear warmup for 1.5K update steps followed by cosine decay, stochastic depth rate 0.2, dropout 0.1, layer-wise learning rate decay 0.95. We evaluate after 160K update steps using a sliding window of 341 pixels and report mIoU over the validation set.

## D.13 $k$-NN CLASSIFICATION AND SEMANTIC SEGMENTATION PROBE PER BLOCK

For the per-block analysis in Figure 3b and Figure 7 we follow the respective settings of $k$-NN classification (Appendix Section D.4) and semantic segmentation linear probing (Appendix Section D.9). The only change is that for semantic segmentation linear probing per-block, we fix the learning rate to 0.1 and only use the patch tokens of the respective block as input to the linear probe.

## D.14 LOSS SCHEDULE VISUALIZATIOS

We visualize the schedules used for scheduling the loss weight of ID heads attached at intermediate blocks in the ablation study (Table 1) in Figure 11.

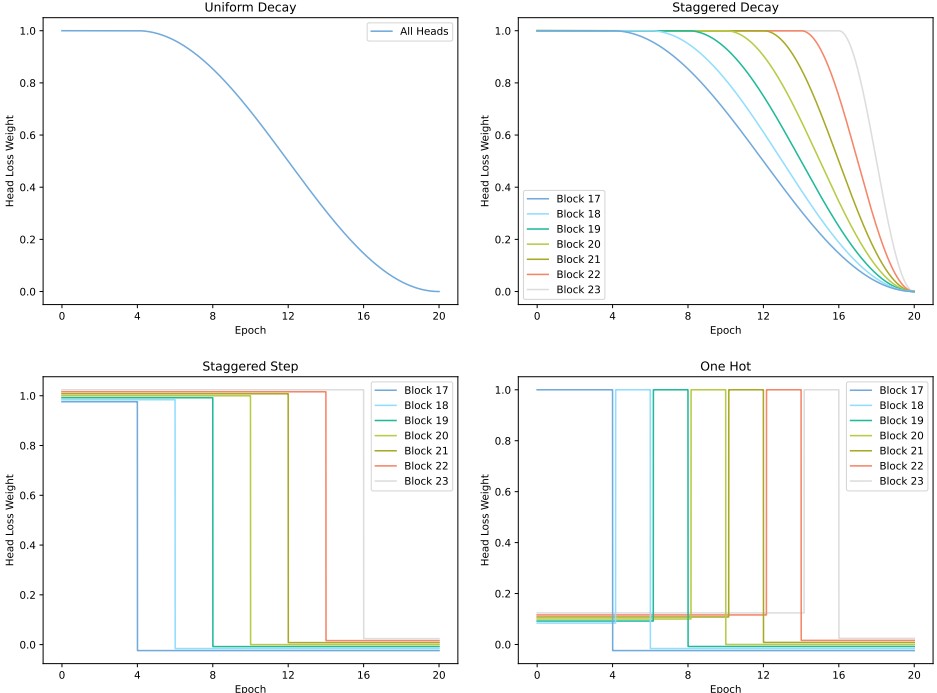

Figure 11: Loss weight schedules for the ablation in Table 1. For visual clarity, small offsets are added when values overlap (for "One Hot" and "Staggered Step").

# E    REPRESENTATION DEGRADATION IN OTHER MODALITIES

We investigate feature degradation of masked pre-trained models from different domains in Figure 12 which show similar trends, suggesting that our methodology could be extended to other modalities.

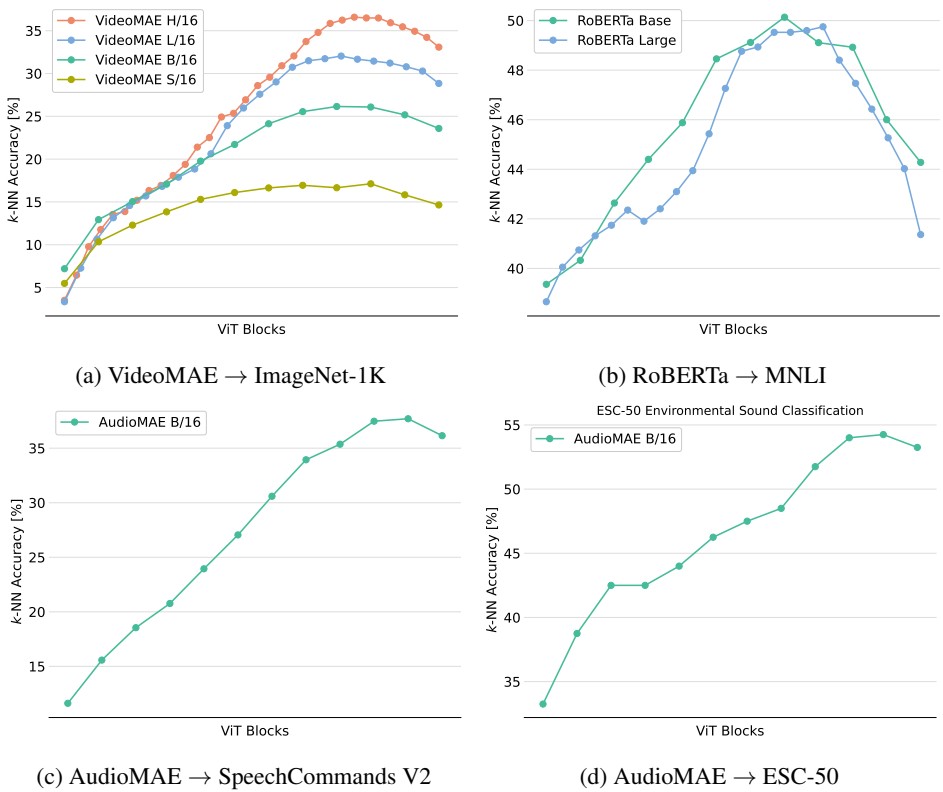

(a) VideoMAE → ImageNet-1K

(b) RoBERTa → MNLI

(c) AudioMAE → SpeechCommands V2

(d) AudioMAE → ESC-50

Figure 12: Feature degradation investigation on other modalities. (a) We evaluate VideoMAE (Tong et al., 2022) on ImageNet-1K classification by treating each image as a single frame video. (b) Entailment classification using RoBERTa (Liu et al., 2019) on the MNLI (Williams et al., 2018) task of the GLUE (Wang et al., 2019a) benchmark. (c) AudioMAE (Huang et al., 2022a) is evaluated on SpeechCommandsV2 (Warden, 2018) audio classification (d) AudioMAE (Huang et al., 2022a) is evaluated on ESC-50 (Piczak) environmental sound classification. Similar trends can be observed in other modalities, often even on smaller models.

# F    RELATION TO NNCLR

Figure 13 shows the difference between NNCLR (Dwibedi et al., 2021) and NNA. NNCLR uses the NN-swap also for the negatives, resulting in a worse signal due to the NNs being retrieved from a FIFO queue of features from previous model states.

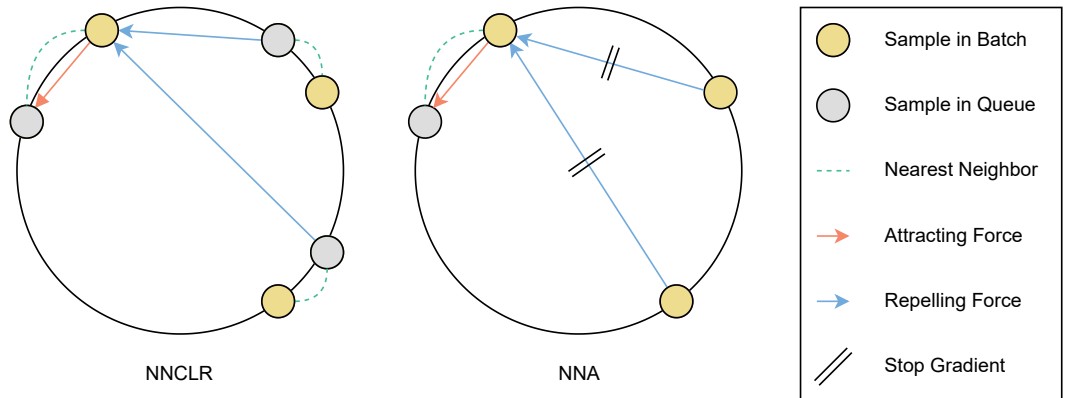

Figure 13: The NN-swap of NNCLR introduces inter-sample correlations between positives but uses features from a previous state of the model. Using the NN-swap only for the positives preserves the inter-sample correlations while using features from the current state of the model as negatives to improve the loss signal.

# G    PRACTITIONER'S GUIDE

We find MIM-Refiner to be easy to tune. By freezing the encoder and training multiple ID heads attached to the encoder with different hyperparameters, one can get a quick and cheap evaluation of suitable hyperparameters. We mainly use two metrics to judge the performance of an ID head:

- Accuracy of a $k$-NN classifier trained on a subset of the data (e.g. 10% of the data). This is relativley cheap to compute and can be done periodically during training. For the $k$-NN classifier, one can use either features of an encoder block or features of intermediate blocks in an ID head to judge the representation at the respective location in the network.

- The accuracy of the NN-swap, i.e. how often is the NN from the NN-swap from the same class as the query sample. This metric is essentially free to compute as the NN-swap is required for training anyways.

