# OpenReview forum: "MIM-Refiner: A Contrastive Learning Boost from Intermediate Pre-Trained Masked Image Modeling Representations"
_ICLR.cc/2025/Conference — ICLR 2025 Poster_

### Official Review · Reviewer_JDMo · 2024-11-02

**Soundness:** 3
**Presentation:** 3
**Contribution:** 2
**Rating:** 6
**Confidence:** 4

**Summary:**

This paper introduces MIM-Refiner, which leverages contrastive learning to boost MIM models. The proposed method is simple and has demonstrated effectiveness in few-shot image classification tasks.

**Strengths:**

S1: This paper is well-written and easy to follow.

S2: This paper is not the first to point out that the encoder in MIM methods partially performs image encoding and representation learning. A similar conclusion is also discussed in [A], highlighting that MIM methods using a single ViT structure tend to face this issue. The reviewer previously conducted experiments on MAE-B, showing that introducing an additional decoder can effectively alleviate this problem. This paper demonstrates that, for methods like MAE that use an asymmetric encoder-decoder architecture， especially in larger models， a small decoder cannot fully decouple encoding and decoding, providing academic insights.

S3: This paper proposes a simple and effective MIM-Refiner method, refining the later blocks of MIM models to enhance MIM representations effectively.

[A] Context Autoencoder for Self-Supervised Representation Learning.

**Weaknesses:**

W1: Existing work [A] has shown that fine-tuning MIM models can enhance their representation capability (for image classification), but the improvement under full fine-tuning is minimal. Additionally, MAE has demonstrated significant transfer performance on dense prediction tasks [B] (object detection/instance segmentation). Fine-tuning MIM models with contrastive learning methods is unlikely to bring substantial improvement and may even negatively impact performance.

W2: Current vision foundation models, such as DINOv2, exhibit strong patch-level representation learning capabilities and combine MIM and CL. Their learned representations have shown effectiveness in tasks like image classification, pixel classification, and depth estimation. Although this paper discusses the relationship between MIM-Refiner and these models, suggesting that MIM-Refiner can build on them for further improvement, I am concerned that MIM-Refiner may degrade pixel-level representation performance for tasks like semantic segmentation or depth estimation (especially when the backbone is fixed).

[A] Layer Grafted Pre-training: Bridging Contrastive Learning And Masked Image Modeling For Label Efficient Representations.

[B] Exploring Plain Vision Transformer Backbones for Object Detection.

**Questions:**

see weaknesses.

---

> ### Author Response · Authors · 2024-11-15
>
> We appreciate your helpful review, particularly the extensive look into related work and are happy to see that our paper was well received and easy to follow. We address your concerns below.
>
> **Conceptual differences to CAE**
>
> The CAE approach is well motivated by their intuition to decouple representation learning from learning the pre-training task where they demonstrate the validity of this intuition via experimental evaluation of their proposed pre-training approach. However, they do not conduct an extensive analysis of pre-trained features or leverage any pre-trained representations, which are two key contributions of our paper. Additionally, we show in Figure 7f of the appendix that CAE also faces degrading feature representation in later blocks, suggesting that their approach alleviates but not fully solves the representation degradation. Contrary, Figure 10b of the appendix shows that refined models achieve peak representation quality in the last block without any feature degradation. We would therefore consider CAE and MIM-Refiner to be orthogonal where CAE presents an improved pre-training method that could also be refined for even better representation. However, as the largest CAE model is a ViT-L/16 and our focus is on large-scale models, we focus on MIM methods that published even larger models.
>
> **Full-finetuning and dense prediction**
>
> Our work builds on the insights of sequential MIM -> ID pre-training methods [1, 2] (as discussed in the related work section) to refine MIM models. These related works do not show improvements or even degradation of full fine-tuning and dense prediction tasks. However, our proposed improvements over previous approaches greatly improve representation quality and, consequently, we do not find our models to suffer from these issues. We show this in Table 5, where our models are on-par or slightly better in full fine-tuning settings with large amounts of data. While we agree that refinement with an ID objective is unlikely to bring major performance gains in full fine-tuning with large amounts of labels, we want to stress that this is the strong suit of MIM models where MIM models show state-of-the-art results. MIM-Refiner therefore heavily improves MIM models in a plethora of benchmarks while preserving (or even slightly improving) upon their state-of-the-art full fine-tuning performances. A visual depiction thereof is provided in Figure 2 left, where MIM-Refiner effectively unifies the advantages of MIM and ID.
>
> Additionally, it has been demonstrated that MAE is exceptionally good at COCO object detection and instance segmentation, where training a pre-trained MAE further via weakly-supervised training on a web-scale dataset of 3 billion images even degraded performance vs a plain ImageNet-1K pre-trained MAE by 0.7 AP [3]. If not even 3B additional images can boost object detection performance, we do not expect a significant performance gain from our extremely short refinement process that does not add additional data.
>
> Nevertheless, we agree that it is important to investigate whether or not the refinement degrades performance on COCO object detection and instance segmentation. We therefore conduct experiments in the setting suggested by reviewer HaAb where we train MAE and MAE-Refined with a Mask R-CNN head using the ViTDet framework on COCO. The results suggest that the refinement process preserves the representation quality also for object detection and instance segmentation downstream tasks. We show results below and added them to the paper (Appendix B.10), together with the above discussion.
>
>
>
> | Model | AP$^\text{box}$ | AP$^\text{mask}$  |
> |---|---|---|
> | MAE | **53.5** | **47.8**  |
> | MAE-Refined | **53.5** | 47.7  |
>
>
> [1] Lehner 2023, "Contrastive Tuning: A Little Help to Make Masked Autoencoders Forget" https://arxiv.org/abs/2304.10520
>
> [2] Jiang 2023, "Layer Grafted Pre-training: Bridging Contrastive Learning And Masked Image Modeling For Label-Efficient Representations" https://openreview.net/forum?id=jwdqNwyREyh
>
> [3] Singh 2023, "The effectiveness of MAE pre-pretraining for billion-scale pretraining" https://arxiv.org/abs/2303.13496

---

> ### Author Response · Authors · 2024-11-15
>
> **Pixel-level representation vs DINOv2**
>
> DINOv2 is an excellent vision foundation model that has demonstrated outstanding performances across various tasks. However, at its core, DINOv2 is a scaled up version of iBOT which uses a dataset of 142M curated dataset of high-quality images. Notable, the dataset is created by retrieving images that are similar to those used for, e.g., semantic segmentation (ADE20K, Cityscapes, Pascal VOC) from an extremely large web-crawled image collection. This dataset curation procedure, together with the patch-level objective of iBOT allows DINOv2 to learn strong pixel-level representations. However, it can not be understated that an essential contributing to this performance is the private web-scale dataset that is highly curated which makes a fair comparison against DINOv2 impossible, as a model pre-trained on 100x more data will obviously outperform a model trained on ImageNet-1K in most cases. The closest comparison to DINOv2 is to compare against the models published by iBOT as their underlying training methodology is identical. We compare against iBOT on all benchmarks where we outperform it by large margins in all settings, including ADE20K segmentation with a frozen encoder (Table 4) and full fine-tuning on ADE20K segmentation (Table 5) using an UperNet semantic segmentation head.
>
> Due to these insights, together with the demonstrated scalability of MIM for billion-scale datasets and model sizes [4], we hypothesize that MIM-Refiner can scale way beyond ImageNet-1K, potentially even outperforming DINOv2 due to highly efficient pre-training. To put it into perspective, [4] trained MAE models up to 6.5B parameters whereas the largest DINOv2 model is 1.1B parameters. MIM-Refiner could leverage the efficient pre-training of MAE to train a 6.5B parameter model followed by a short refinement process on, e.g., the curated DINOv2 dataset which would require a fraction of the compute it would take to train a 6.5B DINOv2 model.
>
>
> [4] Singh 2023, "The effectiveness of MAE pre-pretraining for billion-scale pretraining" https://arxiv.org/abs/2303.13496
>
> **Pixel-level representation preservation under frozen backbone**
>
> We evaluate the performance of MIM models and their refined versions on the pixel-level task of ADE20K semantic segmentation with a frozen encoder in Table 4. The refined models show significant gains in mIoU over their unrefined counterparts across all model sizes. Table 15 in the appendix confirms these results on many more models.

---

### Official Review · Reviewer_HaAb · 2024-11-03

**Soundness:** 3
**Presentation:** 4
**Contribution:** 3
**Rating:** 8
**Confidence:** 4

**Summary:**

---

## **Summary**

The paper identifies the representation degradation issue in Masked Image Modeling (MIM)-pretrained large foundation models. To address this, the authors propose a simple yet effective method to prevent degradation and further improve the representation quality of MIM methods by adding auxiliary contrastive losses to the last layers of Vision Transformers (ViTs) on top of the MIM objective. The paper provides improved performances with large margins over current state-of-the-art (SOTA) methods through extensive experiments and rigorous analysis, demonstrating the success of the proposed approach.

---

**Strengths:**

---

## **Strengths**

- The paper is well-written, with clear observations, a well-developed motivation, a straightforward idea, a clearly-stated method, extensive experiments, and comprehensive analysis.

- It effectively identifies the representational degradation phenomenon in large visual foundation models pre-trained with MIM self-supervised learning (SSL), providing evidence through multiple experiments.

- The proposed method offers a simple and effective solution to prevent this issue and improve the representation quality of MIM SSL.

- Rigorous experiments and analysis are conducted to show the success of the proposed method, with large improvements over current SOTA.

---

**Weaknesses:**

---
### **Limitations**

1. To prevent representation quality degradation in the last layers of ViTs, the authors experiment with contrastive loss, which requires constructing a queue/pool for positive and negative samples. I noticed the proposed method uses a top 20-NN approach to retrieve positive samples in the queue, which could contribute significantly to the increased training time per step. What's the queue size used? how much does it contribute to the increased training time per step?

2. Since the paper emphasizes preserving the richness of representations, evaluation on dense prediction tasks such as object detection and instance segmentation (OD/IS) would be valuable, in addition to the provided segmentation probing on ADE20K.

   - It would be meaningful to compare the performance of MIM-refiner-pretrained ViT-L on COCO object detection against MAE-pretrained ViT-L following the ViTDet framework [1].

   - [1] Li, Y., Mao, H., Girshick, R., & He, K. (2022, October). *Exploring plain vision transformer backbones for object detection.* In European Conference on Computer Vision (pp. 280-296). Cham: Springer Nature Switzerland.

---

### **Recommendation**

Considering the strengths and weaknesses discussed above, my recommendation for this paper is **ACCEPT**. This is a strong paper with a clear contribution.

**Questions:**

---
Since D2V2 is used as a baseline, does the representation degradation issue also appear in the audio and language domains?

---

---

> ### Author Response · Authors · 2024-11-15
>
> Thank you for your valuable review and suggestions to further strengthen the practicality of our approach. We are happy that the storyline of our paper was well understood and appreciated. We respond to your questions below.
>
> **Overhead of queue**
>
> We use a queue size of 65K where, notably, the queue operates within the bottleneck dimension (256 for all model sizes) of the contrastive head.
> The topk NN is found by calculating the cosine similarity for a given sample with all 65K queue entries, and then randomly selecting one of the top k entries in the similarity matrix. Additionally, no gradients flow through the NN-swap so it does not add overhead to the backward pass. These considerations, together with the fact that we train large models lead to a minor overhead from the NN queue. We compare runtimes without a queue, with a top1 NN-swap and with a top20 NN-swap below, which we also included into the paper (Appendix B.17).
>
> | Queue size | topk | L/16 | H/14 | 2B/14  |
> |---|---|---|---|---|
> | 0 | -  | 12.8s | 30.1s | 76.7s  |
> | 65K | 1  | 12.9s | 30.2s | 76.9s  |
> | 65K | 20 | 13.0s | 30.5s | 77.5s  |
>
>
> **Evaluation on dense prediction tasks**
>
> Dense prediction tasks are an important area of computer vision where MIM models have demonstrated exceptional performance. We chose ADE20K semantic segmentation as benchmark for dense downstream tasks as it has established protocols for evaluation via a linear probe (Table 4) and full fine-tuning with feature pyramids and a segmentation head (Table 5) where MIM-Refiner shows strong improvements in linear probing and slight improvements in full fine-tuning. Additionally, it has been demonstrated that MAE is exceptionally good at COCO object detection and instance segmentation, where training a pre-trained MAE further via weakly-supervised training on a web-scale dataset of 3 billion images even degraded performance vs a plain ImageNet-1K pre-trained MAE by 0.7 AP [1]. If not even 3B additional images can boost object detection performance, we do not expect a significant performance gain from our extremely short refinement process that does not introduce additional data.
>
> Nevertheless, we agree that it is important to investigate whether or not the refinement degrades performance on COCO object detection and instance segmentation. We therefore conduct experiments in the suggested setting where we train MAE and MAE-Refined with a Mask R-CNN head using the ViTDet framework on COCO. The results suggest that the refinement process preserves the representation quality also for object detection and instance segmentation downstream tasks. We show results below and added them to the paper (Appendix B.10), together with the above discussion.
>
>
>
> | Model | AP$^\text{box}$ | AP$^\text{mask}$  |
> |---|---|---|
> | MAE | **53.5** | **47.8**  |
> | MAE-Refined | **53.5** | 47.7  |
>
>
> [1] Singh 2023, "The effectiveness of MAE pre-pretraining for billion-scale pretraining" https://arxiv.org/abs/2303.13496
>
>
> **Representation degradation in other modalities**
>
> We find it an interesting avenue to explore and will follow-up shortly, as we do not have a setup for the other D2V2 modalities ready-to-go and focused on getting timely object detection result to facilitate discussion.
>
> However, we did have a pipeline for VideoMAE [2] and AudioMAE [3] ready-to-go, which shows a similar trend, interestingly enough, also for smaller models. We include these preliminary result Appendix E.
>
> [2] Tong 2022, "VideoMAE: Masked Autoencoders are Data-Efficient Learners for Self-Supervised Video Pre-Training" https://arxiv.org/abs/2203.12602
>
> [3] Huang 2022, "Masked Autoencoders that Listen" https://arxiv.org/abs/2207.06405

---

> ### Author Response · Authors · 2024-11-19
>
> **Representation degradation in other modalities**
>
> We updated the paper to also include results for models pre-trained with masked language modeling. As D2V2 only trains a single model size (ViT-B) for its language modeling experiments, we instead opt for RoBERTa models which train models up to a ViT-L (and are also pre-trained with a masked language modeling objective). Our analysis now covers 3 different domains (Video, Language, Audio) where feature degradation is present in all modalities.

---

### Official Review · Reviewer_BbVs · 2024-11-03

**Soundness:** 3
**Presentation:** 3
**Contribution:** 3
**Rating:** 6
**Confidence:** 4

**Summary:**

This paper presents a contrastive learning boosting method called MIM-Refiner to refine the features of pre-trained MIM models. MIM-Refiner leverages multiple instance discrimination heads (ID) which are connected to different immediate layers. Each ID head contains a contrastive loss that captures semantic information to improve the quality of learned representations. By training a few epochs, the features of MIM-Refiner surpass the current MIM models on multiple experiments: on ViT-H with data2vec 2.0 on ImageNet-1K, the accuracy of the proposed method reaches state-of-the-art performance on linear probing and low-shot classification.

**Strengths:**

1. This paper proposes a detailed analysis of the blocks of MIM models in which different blocks extract features with a specific focus and the most efficient features learned by MIM are from the middle blocks.

2. A contrastive learning-based method called MIM-Refiner is proposed to refine the representation of current MIM models by attaching the middle layers with ID objective.

3. Experimental results show the effectiveness and generalization ability of MIM-Refiner on downstream tasks.

**Weaknesses:**

1. As the discussion of end-to-end training, the proposed method MIM-Refiner seems to be a two-step training method, with first step training MIM models and fine-tuning the updated models by incorporating ID heads to middle layers. Practically, this might increase the complexity of the training paradigm and deployment. Is it possible to improve the proposed method with end-to-end training on MIM and ID? If not, what are the potential bottlenecks to circumvent this goal?

2. There is no overview diagram that shows the detailed architecture of MIM-Refiner or how the training diagram goes. The diagram in Figure 4 provides partial information but does not clearly illustrate these points.

**Questions:**

Please refer to weakness.

---

> ### Author Response · Authors · 2024-11-15
>
> Thank you for your review and helpful comments to help us improve the paper. We address your points individually.
>
> **End-to-end training of ID and MIM**
>
> MIM and ID objectives are quite conflicting where, roughly speaking, MIM considers every pixel/patch to be equally important, as the whole image needs to be reconstructed, while ID only cares about distinguishing positive and negatives in a batch of samples, which implicitly weights pixels/patches by their information content. This conflict also results in vastly different hyperparameter choices that are required for optimal performance. For example, MIM models use little image augmentations (only cropping and resizing but no color augmentations) but high masking ratios (75\%). Contrary, ID models use sophisticated image augmentation pipelines with different augmentation strengths per view, color augmentations and multi-crop augmentation but only small masking ratios (e.g, 25\% for iBOT/DINOv2). These findings were highlighted by two related works [1, 2] (as cited in Section 5.2) which independently came to the conclusion that a sequential training can effectively alleviate this conflict.
>
> Also in terms of scalability, MIM models scale extremely well to large model sizes, whereas ID models require much more compute and data. As larger models typically perform better, it is desirable to develop scalable approaches. Attempts to develop end-to-end combinations of MIM and ID introduce large compute overheads due to, e.g., target networks or lower masking ratios, which heavily limits scalability. This is evident by the fact that these methods have not been trained on model scales beyond ViT-L, where the largest model of most methods is a ViT-B. Prominent examples are [3, 4, 5]. In contrast, our sequential approach can effortlessly scale up to a 2B parameter model, on a relatively small compute budget, by leveraging the compute efficiency of MIM models.
>
> We also show that end-to-end MIM and ID combinations do not fully leverage the potential of the ID objective in Appendix B.15, where refining a CMAE-B model with our proposed methodology also improves performance despite the fact that an ID objective was already used in combination with the MIM objective in pre-training.
>
> Additionally, MIM-Refiner models have strong off-the-shelf performances, so while MIM-Refiner requires a multi-stage pre-training, it makes downstream training much easier where simple models like a $k$-NN classifier or a linear probe reach somewhat comparable results to fully fine-tuning a model. For example, D2V2-Refined-H achieves 84.7\% with a simple linear probe on ImageNet-1K classification that can be easily trained on a single GPU. Fine-tuning the same model boosts this performance by 2.1 \% while requiring much more compute, necessitating multi-GPU or even multi-node training setups.
>
> To summarize, the fundamental differences of MIM and ID require trade-offs in an end-to-end setting and we therefore opt for a sequential approach. While this increases the complexity of the training pipeline, multi-stage pre-training pipelines are somewhat common, and, since pre-training has to be done only once before a model can be fine-tuned (or simply evaluated) on a broad range of downstream tasks, multi-stage pre-training pipelines do not drastically decrease practicality. For example, in language modeling (e.g. [6]), it is very common to "refine" models after pre-training to integrate alignment with human preferences, reasoning, long context understanding, tool usage or safety guards. These multi-stage pre-training pipelines in language models are common because their benefit outweighs the added complexity. We would argue that this also holds true for MIM-Refiner, where the benefits of broader adaptability to many more use cases outweighs the additional pre-training pipeline complexity.
>
>
> [1] Lehner 2023, "Contrastive Tuning: A Little Help to Make Masked Autoencoders Forget" https://arxiv.org/abs/2304.10520
>
> [2] Jiang 2023, "Layer Grafted Pre-training: Bridging Contrastive Learning And Masked Image Modeling For Label-Efficient Representations" https://openreview.net/forum?id=jwdqNwyREyh
>
> [3] Huang 2022, "Contrastive Masked Autoencoders are Stronger Vision Learners" https://arxiv.org/abs/2207.13532
>
> [4] Zhou 2021, "iBOT: Image BERT Pre-Training with Online Tokenizer" https://arxiv.org/abs/2111.07832
>
> [5] Assran 2022, "Masked Siamese Networks for Label-Efficient Learning" https://arxiv.org/abs/2204.07141
>
> [6] Dubey 2024, "The Llama 3 Herd of Models" https://arxiv.org/abs/2407.21783

---

> ### Author Response · Authors · 2024-11-15
>
> **Clarification of training diagram in Figure 4**
>
> We aim to depict the training pipeline of MIM-Refiner in Figure 4 by showing the differences between ID, MIM and MIM-Refiner with the goal that the "copy" lines should indicate that MIM-Refiner starts from a pre-trained MIM model where multiple ID heads are attached at later blocks instead of only a single ID head at the last block.
>
> Additionally, due to reviewer Auz1's feedback, we added more details to the introductory paragraph of Section 4, which hopefully also clarifies the overall training methodology if it was not clear from Figure 4.
>
> We regret that this was not found to illustrate our method clearly. We hope the additional information clarifies our experimental setup and otherwise would be keen to hear more details about what information is missing or was not clearly presented in the figure.

---

### Official Review · Reviewer_Auz1 · 2024-11-04

**Soundness:** 3
**Presentation:** 3
**Contribution:** 3
**Rating:** 6
**Confidence:** 4

**Summary:**

The paper focuses on bridging the gap between large MIM pre-trained models and SOTA methods. The paper first discovers that  MIM models have different types of blocks: those that mainly improve the pre-training objective and others that are responsible for abstraction. Then, the paper proposes a method MIM-Refiner, which adds Instance Discriminator heads on pre-trained MIM models for refinement. The ID heads exploit the intermediate representations to consistently improve the performance of MIM pretrained models. While the performance gains on large dataset full-finetuning are small, the proposed methods show remarkable gains on few-shot settings.

**Strengths:**

1. The paper first points out the influence of the lightweight decoder on the feature learning of the encoder in MIM methods.
2. The analyzing part is well-written.

**Weaknesses:**

1. The description of the method and experimental setup needs to be clarified. (a) Which blocks need to be fine-tuned during refinement, or do all blocks need to be fine-tuned? (b) How many epochs are needed to refine different models? (c) What is the structure of the ID head? Answers to all these questions should be contained in the manuscript.
2. Unfair comparison. The paper misses an important baseline - train the original model with 0 heads with the same epochs to demonstrate the importance of refinement (instead of just training more epochs).
3. Some typos. L267-269, see Table 1 instead of Figure 3b.

**Questions:**

Please refer to the weakness. I believe a clear description of the method and experimental setup is one of the most important things when writing a paper (weakness 1).

Additional question: what does the “relative” in Figure 3(d) mean? Does the value calculated by the performance of ( the i+1 th layer - the i-th layer)?

---

> ### Author Response · Authors · 2024-11-15
>
> Thank you for your profound review and suggestions that helped us to improve our paper. We are glad that the analysis part was found to be a nice read. We corrected the typos and address your points below.
>
> **Clarifications on experimental setup**
>
> We agree that the most important hyperparameters should be presented in the main text. Therefore, we restructured the beginning of Section 4 to include an overview of the experimental setting. Your comment also made us aware that we never referred to the full set of implementation details, as included in Appendix C, including how many blocks are finetuned (C.5 Table 22), how many epochs are needed for different models (C.5 Table 22) or the ID head structure (C.4).
> We fixed this oversight by appropriately referring to it in Section 4.
>
>
> **Comparison against prolonged MIM training**
>
> We provide a comparison similar to the proposed one already within our paper, where CrossMAE-L is pre-trained for only 800 epochs (due to computational resource restrictions of the authors), while MAE-L is pre-trained for 1600 epochs. One can clearly see in, e.g., Table 10 or Table 14 of the appendix that CrossMAE-Refined outperforms MAE, where CrossMAE-Refined is pre-trained for 800 epochs followed by 30 epochs of refinement while MAE is pre-trained for 1600 epochs.
>
> While we see that this is not a perfect comparison due to the differences in the decoder between MAE and CrossMAE, we believe that it provides sufficient evidence to underline the effectiveness of our method. Unfortunately, a direct comparison by prolonging the training of a pre-trained MIM model has multiple issues as we will outline below.
>
> First, MIM models are commonly pre-trained using a cosine annealing learning rate schedule, which means that for a proper comparison, one would need to train the whole model from scratch with a higher epoch count, which is extremely expensive (note that we simply downloaded pre-trained MIM checkpoints and never needed to train one from scratch). During the later epochs of the cosine annealing schedule, the model is updated with tiny learning rates. If one would use the pre-trained model and start training it for some more epochs, using a standard warmup into cosine annealing schedule and the same pre-training objective, it would essentially destroy the updates of the last few epochs due to increasing learning rate, before conducting the "same" updates again once the learning rate decreases again.
>
> Second, often times, only the ViT encoder is published (without the decoder), so prolonging the training would require some sort of mechanism to initialize the decoder as starting the training with a randomly initialized decoder would most likely degrade the encoder features until the decoder has learned to produce sensible reconstructions.
>
> Third, MIM models are often trained for many epochs. For example, MAE models are trained for 1600 epochs. It is highly unlikely that prolonging this training by, e.g., another 50 epochs, will significantly change the performance of the model as performance gains tend to saturate.
>
> Lastly, the number of epochs for MIM pre-training is often optimized as a hyperparameters. For example, in data2vec 2.0 the amount of epochs is decreased with model size, which suggests that larger models converge faster. Therefore, adding additional epochs could also degrade performance due to overfitting. While compute costs could be another explanation why the number of epochs was decreased with model size, we find the overfitting explanation more realistic as D2V2 is extremely compute efficient and D2V2-B or D2V2-L could have easily been trained longer as their compute budget also included training of a D2V2-H model. Therefore, prolonged training of smaller models would not have significantly impacted the total compute budget.
>
>
> **Clarification on "relative" in Figure 3d**
>
> Your conclusion is correct, we calculate the relative improvement by subtracting the performance of block i + 1 from the performance of block i. Additionally, we divide by the maximum relative improvement to put both performance metrics (k-NN accuracy and reconstruction loss) into the same value range with 1 as upper bound. We included the methodology to calculate the relative improvement into the caption of Figure 3 and expanded the description in Appendix D.2.

---

### Author Response · Authors · 2024-11-15
**General Response**

We thank all reviewers for their positive feedback, constructive comments and suggestions.
We are pleased to see that the reviewers highlighted the clarity of our paper and appreciated our analysis as well as our extensive experiments. Several reviewers recognized the clear motivation of our approach from the thorough analysis.

We updated the paper to incorporate the feedback of the reviewers. To summarize, we made the following changes:

- Added more details to the experimental setup in the main paper (first paragraph in Section 4) such as training duration and included a reference to the extensive implementation details and hyperparameters in the appendix (reviewer Auz1).
- Included methodology to calculate the relative improvement in Figure 3d also in the caption instead of only in the appendix (reviewer Auz1).
- Added results for COCO object detection and instance segmentation in Appendix B.10 (reviewer HaAb and JDMo)
- Added preliminary analysis of feature degradation of masked pre-training in other modalities in Appendix E (reviewer HaAb).
- corrected various typos

Additionally, we respond to each review individually, addressing the raised questions and concerns.

---

> ### Author Response · Authors · 2024-11-19
>
> We updated the paper to include an analysis for feature degradation for models of different modalities that are pre-trained by reconstructing a part of the masked input in Appendix E. Our analysis covers 3 different domains (Video, Language, Audio) where feature degradation is present in all modalities.

---

### Meta-Review · Area_Chair_Xp3v · 2024-12-17

**Metareview:**

After discussion, this submission received 4 positive scores . The major concerns about the experimental details,  preliminary analysis  and method clarity were comprehensively solved. After reading the paper, the review comments and the rebuttal, the AC thinks the remaining issue is to include all the revised content to the camera-ready version and correct typographical errors.

**Additional Comments On Reviewer Discussion:**

After discussion, all the reviewers gave the positive scores and did not raise further concerns or issues.

---

### Decision · Program_Chairs · 2025-01-22

Accept (Poster)